# Softmax as Linear Attention in the Large-Prompt Regime:
# a Measure-based Perspective

**Etienne Boursier** [1]  **Claire Boyer** [1 2]

## Abstract

Softmax attention is a central component of transformer architectures, yet its nonlinear structure poses significant challenges for theoretical analysis. We develop a unified, measure-based framework for studying single-layer softmax attention under both finite and infinite prompts. For i.i.d. Gaussian inputs, we lean on the fact that the softmax operator converges in the infinite-prompt limit to a linear operator acting on the underlying input-token measure. Building on this insight, we establish non-asymptotic concentration bounds for the output and gradient of softmax attention, quantifying how rapidly the finite-prompt model approaches its infinite-prompt counterpart, and prove that this concentration remains stable along the entire training trajectory in general in-context learning settings with sub-Gaussian tokens. In the case of in-context linear regression, we use the tractable infinite-prompt dynamics to analyze training at finite prompt length. Our results allow optimization analyses developed for linear attention to transfer directly to softmax attention when prompts are sufficiently long, showing that large-prompt softmax attention inherits the analytical structure of its linear counterpart. This, in turn, provides a principled and broadly applicable toolkit for studying the training dynamics and statistical behavior of softmax attention layers in large prompt regimes.

[1]Université Paris-Saclay, CNRS, Inria, Laboratoire de mathématiques d'Orsay, 91405, Orsay, France [2]Institut Universitaire de France. Correspondence to: Etienne Boursier <etienne.boursier@inria.fr>.

*Proceedings of the 43rd International Conference on Machine Learning*, Seoul, South Korea. PMLR 306, 2026. Copyright 2026 by the author(s).

## 1. Introduction

Modern transformer architectures rely fundamentally on attention mechanisms (Vaswani et al., 2017), whose ability to extract and combine information from long prompts underpins the remarkable empirical success of large language models and related systems. The nonlinear structure of the softmax attention mechanism lies at the heart of the empirical success of transformer architectures, yet it also presents a major obstacle to theoretical analysis. Even in the single-layer setting, the dependence of softmax attention on all pairwise token interactions makes its behavior difficult to characterize, both statistically and from an optimization standpoint. To circumvent these challenges, a substantial body of recent theoretical work has turned to linear attention as a proxy model (Ahn et al., 2024), whose algebraic simplicity enables explicit analyses of learning dynamics and generalization. However, despite the widespread use of linear attention in theoretical studies, it is well documented that softmax attention exhibits significantly stronger empirical performance, particularly in terms of length generalization, expressivity and its ability to extract task structure from long prompts (He et al., 2025; Duranthon et al., 2025; Dragutinović et al., 2025). Strikingly, from a theoretical perspective, it is still not understood whether softmax attention should systematically outperform, or even consistently match, the capabilities of its linear counterpart. This gap between empirical evidence and theoretical understanding motivates the need for a framework that can relate the behavior of softmax and linear attention in a principled manner.

To address this gap, we rely on a unifying framework based on a measure-based representation of attention layers (Vuckovic et al., 2020; Sander et al., 2022). This perspective allows us to systematically compare the finite-prompt model with its infinite-prompt counterpart and to identify regimes in which softmax attention behaves effectively linearly with respect to the input-token distribution. Such a perspective opens the door to simplified optimization analyses in the infinite-prompt limit, which can then be translated back into realistic settings involving softmax attention.

**Contributions.** We consider a unified, measure-based framework, notably developed by Vuckovic et al. (2020); Sander et al. (2022), that allows to analyze single-layer transformers with both finite and infinite prompts. A key observation in this setting is that, for i.i.d. Gaussian input tokens, the action of a softmax attention layer over an infinite number of tokens becomes linear with respect to the underlying input-token measure. Building on this insight, our main contributions are as follows:

- **Concentration toward the infinite-prompt limit.** We first establish non-asymptotic concentration bounds for both the output and the gradient of a softmax attention layer as the prompt length grows. These results quantify how rapidly the behavior of a finite-prompt layer approaches its infinite-prompt limit.

- **Stability of concentration along training dynamics.** While the previous bounds hold pointwise for fixed attention parameters, we further show—within a general in-context learning setting—that this concentration persists along the entire training trajectory of the attention layer. Informally we show that for a general in-context learning task,

$$\lim_{t\to\infty} \mathcal{R}(\theta_L(t)) \le \lim_{t\to\infty} \mathcal{R}(\theta_\infty(t)) + o_{L\to\infty}(1),$$

where $\mathcal{R}$ is the theoretical risk associated to the in-context learning task and $\theta_L$ (resp. $\theta_\infty$) are the model parameters obtained by following the gradient flow for the theoretical risk minimization with a finite (resp. infinite) prompt length $L$. This, in turn, enables us to approximate the training dynamics at finite prompt length by those of the infinite-prompt model. The latter is typically much easier to analyze from an optimization standpoint.

- **Leveraging the infinite-prompt limit: softmax attention solves in-context linear regression.** We then leverage our previous results and show that they apply in the setting of in-context learning for linear regression with Gaussian inputs. In this regime, our analysis allows the linear-attention results of Zhang et al. (2024) to transfer directly to softmax-attention dynamics, provided that the prompt length is sufficiently large. More broadly, under centered Gaussian inputs, our framework extends theoretical analyses developed for linear attention to softmax attention in the large-prompt regime.

**Related works.** Measure-based formulations recently yielded seminal insights into gradient-based analyses of one-hidden-layer networks (Nitanda and Suzuki, 2017; Mei et al., 2018; Rotskoff and Vanden-Eijnden, 2018; Bach and Chizat, 2022). While these works interpret the model parameters as measures, Pevny and Kovarik (2019); De Bie et al. (2019); Zweig and Bruna (2021)

adopted a different perspective by considering architectures that take measures as inputs. This viewpoint has become particularly relevant with the emergence of the attention mechanism, and has since been explored in numerous works (Vuckovic et al., 2020; Sander et al., 2022; Geshkovski et al., 2025; Bruno et al., 2025; Burger et al., 2025; Zimin et al., 2025; Castin et al., 2025). These studies mostly focus on inference through attention layers with fixed weights. Our work further develops this line of research by adopting a measure-based representation of the input prompt while studying the training dynamics of a single attention layer in the regime of large but finite prompt length. The concurrent work of Bohbot et al. (2025) also investigates the concentration of the attention output in the large-prompt regime, similarly to the results presented in Section 3. Although their analysis yields sharper convergence rates, it is restricted to the concentration of moments of the output distribution. In contrast, our results establish $L_2$ concentration for both the attention output and the gradients with respect to the trainable parameters. This latter property is essential for deriving the training dynamics results developed in the subsequent sections.

In addition, a central phenomenon identified in the behavior of transformers is in-context learning (ICL, Brown et al., 2020; Olsson et al., 2022; Min et al., 2022; Chan et al., 2022). It refers to the ability of a model to implement new tasks directly from the prompt at inference time, without modifying its weights or running any additional optimization. Early empirical and mechanistic studies (Garg et al., 2022; Raventós et al., 2023) show that, when trained on prompts encoding linear regression tasks, transformers implicitly implement in-context linear regression within their attention layers. This observation has motivated a series of theoretical analyses focusing on linear attention mechanisms in a variety of settings (Von Oswald et al., 2023; Ahn et al., 2023; Mahankali et al., 2024; Zhang et al., 2024; Lu et al., 2025). Notably we build on the work of Zhang et al. (2025), which establishes precise learning dynamics and generalization behaviors for in-context linear regression with linear attention mechanism.

Beyond linear attention, several works have begun to investigate softmax attention for in-context linear regression. Huang et al. (2024); He et al. (2025) provide analyses of the training dynamics, though respectively under restrictive data assumptions and under specific, nonstandard optimization algorithms. Of particular relevance to our work, Chen et al. (2024) show that softmax attention can learn multi-task linear regression under Gaussian covariates, which may be viewed as a refined and more quantitative counterpart of our Theorem 5.2. Their analysis, however, relies on heavily technical arguments and is restricted to the case of isotropic covariates, a setting that substantially simplifies both the dynamics and their analysis (see,

e.g., He et al., 2025, Section 5.3). In contrast, our work handles anisotropic covariates and introduces a more compact and user-friendly proof strategy, leading to a general toolkit for analyzing softmax attention layers.

**Outline.** Section 2 recalls the measure-based formalism underlying our view of attention layers. Section 3 establishes concentration bounds showing how finite-prompt softmax attention layers (and related functions) concentrate around their linear infinite-prompt counterparts. Building on these results, Section 4 demonstrates how theoretical analyses of training dynamics for linear attention can be transferred to the softmax setting in general in-context learning problems. Section 5 specializes this analysis to the case of in-context linear regression. Finally, Section 6 illustrates our different results on numerical toy experiments.

## 2. Measure based view of attention

In this section, we recall the measure-based formulation of attention introduced, for instance, by Vuckovic et al. (2020); Sander et al. (2022); Castin et al. (2025). In this framework, the transformer receives both a prompt of $d$-dimensional tokens, represented as a probability measure $\mu \in \mathcal{P}(\mathbb{R}^d)$, and a query token $z \in \mathbb{R}^d$. We denote by $T^{K,Q,V}$ a transformer equipped with a softmax self-attention layer, parameterized by key, query, and value matrices $K, Q, V \in \mathbb{R}^{d \times d}$. This formulation allows us to treat finite- and infinite-length prompts within a unified framework.

**Finite-length prompt.** We begin by considering the case where the prompt consists of a finite sequence of $L$ tokens $(z_1, \ldots, z_L) \in \mathbb{R}^{d \times L}$. In this case, the output of an attention layer consists of $L$ transformed tokens, each of which is computed independently of the original token order—once positional encodings and masking are ignored. This permutation equivariance naturally leads us to represent the prompt as the empirical measure $\hat{\mu}_L = \frac{1}{L} \sum_{\ell=1}^{L} \delta_{z_\ell}$ associated with the tokens $(z_1, \ldots, z_L)$.

The transformation operated by a self-attention layer $T^{K,Q,V}$ can then be seen as the following map

$$T^{K,Q,V} : \mathcal{P}(\mathbb{R}^d) \times \mathbb{R}^d \rightarrow \mathbb{R}^d$$
$$(\mu, z) \mapsto T^{K,Q,V}[\mu](z)$$

with

$$T^{K,Q,V}[\mu](z) := \frac{\int_{\mathbb{R}^d} \exp(\langle Qz, Kz' \rangle) V z' \mathrm{d}\mu(z')}{\int_{\mathbb{R}^d} \exp(\langle Qz, Kz' \rangle) \mathrm{d}\mu(z')},$$

where, in the case of finite-length prompt with $\hat{\mu}_L = \frac{1}{L} \sum_{\ell=1}^{L} \delta_{z_\ell}$, the $\ell$-th transformed token is given by

$$T^{K,Q,V}[\hat{\mu}_L](z_\ell) = \frac{\sum_{k=1}^{L} \exp(\langle Qz_\ell, Kz_k \rangle) V z_k}{\sum_{k=1}^{L} \exp(\langle Qz_\ell, Kz_k \rangle)},$$

which coincides with the classical definition of self-attention.

**Infinite-length prompt.** When the input tokens are independently and identically distributed according to $\mu$, and the prompt length $L$ tends to infinity, the input prompt can be effectively viewed as the distribution $\mu$ itself. Indeed, one can observe, by the strong law of large numbers, that for any $z \in \mathbb{R}^d$,

$$T^{K,Q,V}[\hat{\mu}_L](z) \xrightarrow[L \to +\infty]{\text{a.s.}} T^{K,Q,V}[\mu](z).$$

This limit case is particularly interesting when the distribution $\mu$ of the input tokens is a Gaussian measure. Indeed, the transformation applied by $T^{K,Q,V}$ then preserves the Gaussian nature of the distribution. More precisely, it corresponds to an affine transformation of the input, as formalized in the following lemma.

**Lemma 2.1** (Castin et al. 2025, Lemma 4.1). *Assume that $\mu$ is a Gaussian measure $\mathcal{N}(m, \Gamma)$ of mean $m \in \mathbb{R}^d$ and covariance matrix $\Gamma \in \mathbb{R}^{d \times d}$. Then, the output obtained from the attention layer $T^{K,Q,V}$ is characterized, for any $z \in \mathbb{R}^d$, as*

$$T^{K,Q,V}[\mu](z) = Vm + V\Gamma K^\top Qz.$$

*In particular, the pushforward measure $T^{K,Q,V}_{\#}\mu$ of $\mu$ through $T^{K,Q,V}$ is Gaussian, i.e., $T^{K,Q,V}_{\#}\mu = \mathcal{N}(V(\mathrm{I}_d + \Gamma K^\top Q)m, V\Gamma K^\top Q\Gamma(V\Gamma K^\top Q)^\top)$.*

This result states that when the full token measure $\mu$ is provided as prompt input, softmax-based attention behaves essentially like a linear transformation. More precisely, it coincides with a normalized self-attention layer for centered Gaussian inputs (see Appendix B.2 for further details).

However, it remains unclear how this relationship extends to realistic settings with finite—albeit possibly large—prompt lengths $L$. To address this question, Section 3 provides quantitative bounds characterizing the concentration of $T^{K,Q,V}[\hat{\mu}_L]$ towards its infinite-prompt counterpart $T^{K,Q,V}[\mu]$, when $\hat{\mu}_L$ is the empirical measure of $L$ i.i.d. tokens distributed according to $\mu$. Notably, these concentration results are derived in a setting that goes beyond Gaussian measures, allowing for more general input distributions.

## 3. Concentration of softmax attention

In this section, we draw a connection between a softmax-based attention layer with a finite-length prompt and an attention layer that would have received an infinite number of input tokens, i.e., the full token distribution.

**Proposition 3.1** (Output concentration bound). *Assume that $\mu$ and $\nu$ are respectively $\sigma$ and $1$ centered sub-*

*Gaussian measures with $\sigma \geq 1$. Then there exist constants $c_1, c_2 > 0$ depending solely on $d, V, K, Q$ such that,*

$$\mathbb{E}\left[\left\|T^{K,Q,V}[\hat{\mu}_L] - T^{K,Q,V}[\mu]\right\|^2_{L^2(\nu)}\right] \leq c_1 \sigma^6 \frac{\ln(L)}{L^{c_2/\sigma^2}} \tag{1}$$

*where the expectation is taken over $(z_1, \ldots, z_L) \sim \mu^{\otimes L}$ with $\hat{\mu}_L = \frac{1}{L}\sum_{\ell=1}^{L} \delta_{z_\ell}$.*

**Sketch of proof.** Although the input tokens are sub-Gaussian, the proof of Proposition 3.1 does not follow directly from standard sub-Gaussian concentration arguments on the sums appearing in the numerator and denominator of $T^{K,Q,V}[\hat{\mu}_L](z)$. Indeed, the softmax weights involve potentially heavy-tailed random variables of the form $\exp(z_k^\top K^\top Q z)$. On the high-probability event where these terms remain uniformly bounded, we apply independent concentration inequalities to the numerator and denominator of the softmax expression. In contrast, on the complementary (low-probability) event where one of the terms $\exp(z_k^\top K^\top Q z)$ becomes exceptionally large, we exploit the fact that the attention output lies in the convex hull $\mathrm{Conv}(z_1, \ldots, z_L)$. As a consequence, its norm is bounded by $\max_k \|V z_k\|$, which can be controlled using standard maximal inequalities for sub-Gaussian random variables. Extending these arguments to bounds that also average over the query token $z$ introduces additional challenges, as concentration estimates obtained for fixed $z$ typically deteriorate with $\|z\|$. To address this issue, we again adopt a case-based analysis, employing different bounding techniques depending on whether $\|z\| \gtrsim \sqrt{d}$ or $\|z\| \lesssim \sqrt{d}$. □

Precise dependencies in $d, V, K, Q$ can be deduced from Equation (12) in the Appendix.

**Remark 3.2** (Handling the autocorrelation in attention). *In what follows, the analysis relies on concentration arguments of*

$$\left\|T^{K,Q,V}[\hat{\mu}_L] - T^{K,Q,V}[\mu]\right\|^2_{L^2(\nu)} =$$
$$\mathbb{E}_{z\sim\nu}\left[\left\|T^{K,Q,V}[\hat{\mu}_L](z) - T^{K,Q,V}[\mu](z)\right\|^2\right],$$

*(and some of its variants), as obtained in Proposition 3.1. Here the expectation is taken with respect to a fresh independent variable $z \sim \nu$. A careful reader may notice that this differs from the practical setting of attention layers, where the model is typically evaluated on tokens that also appear in the prompt. In that practical context, the query token is part of the empirical measure used to build $\hat{\mu}_L$. A mathematical faithful formulation would therefore map a new query token $z$ to $T^{K,Q,V}\left[\frac{L}{L+1}\hat{\mu}_L + \frac{1}{L+1}\delta_z\right](z)$, where we explicitly include the contribution of the query token in the empirical measure. A rigorous adaptation of*

*our proofs to this convention does not introduce any new technical difficulties (although it makes the arguments less readable), since both $\hat{\mu}_L$ and $\frac{L}{L+1}\hat{\mu}_L + \frac{1}{L+1}\delta_z$ converge to $\mu$ at the same rate.*

The concentration phenomenon obtained in Proposition 3.1 extends to related quantities associated with the attention layer, most notably the gradients with respect to the attention parameters. This extension is particularly important for the analysis of training dynamics in attention-based models. To streamline the exposition, we henceforth focus on the dynamics with respect to $U = K^\top Q$ and $V$, so that the measure-based representation of an attention layer becomes

$$T^{U,V}[\mu] : z \mapsto \frac{\int_{\mathbb{R}^d} \exp(z'^\top U z) V z' \mathrm{d}\mu(z')}{\int_{\mathbb{R}^d} \exp(z'^\top U z) \mathrm{d}\mu(z')}.$$

Such a parametrization is common in optimization analyses of attention architectures (see e.g., Ahn et al., 2023; Zhang et al., 2024; Lu et al., 2025; Zhang et al., 2025). Leaning on this reparameterization, we provide in the following concentration bounds for $\nabla_U T^{U,V}$ and $\nabla_V T^{U,V}$.

**Assumption 3.3.** *The distributions $\mu$ and $\nu$ are respectively $\sigma$ and $1$ sub-Gaussians and for any $p, p' \in \{0, 4, 8\}$, there exist constants $M_{p,p'} \in \mathbb{R}$ such that*

$$\mathbb{E}_{Z\sim\nu}\left[\frac{\mathbb{E}_{Z_1\sim\mu}[\exp(Z_1^\top U Z)\|Z_1\|^p\|Z\|^{p'}]}{\mathbb{E}_{Z_1\sim\mu}[\exp(Z_1^\top U Z)]}\right] \leq \sigma^{2p} M_{p,p'},$$

*where $M_{p,p'}$ potentially depends on $U$.*

The moment conditions of Assumption 3.3 are typically satisfied for bounded and Gaussian distributions.

**Proposition 3.4** (Gradient concentration bound). *Assume that $\mu$ and $\nu$ are respectively $\sigma$ and $1$ centered sub-Gaussian measures with $\sigma \geq 1$ and that Assumption 3.3 holds. Then there exist constants $c_1, c_2 > 0$ depending solely on $d, V, U, (M_{p,p'})_{p,p'}$ such that,*

$$\mathbb{E}\left[\left\|\nabla_V T^{U,V}[\hat{\mu}_L] - \nabla_V T^{U,V}[\mu]\right\|^2_{L^2(\nu)}\right] \leq c_1 \sigma^6 \frac{\ln(L)}{L^{c_2/\sigma^2}}$$

$$\mathbb{E}\left[\left\|\nabla_U T^{U,V}[\hat{\mu}_L] - \nabla_U T^{U,V}[\mu]\right\|^2_{L^2(\nu)}\right] \leq c_1 \sigma^{12} \frac{\ln(L)^2}{L^{c_2/\sigma^2}}$$

*where the expectation is taken over $(z_1, \ldots, z_L) \sim \mu^{\otimes L}$ with $\hat{\mu}_L = \frac{1}{L}\sum_{\ell=1}^{L} \delta_{z_\ell}$.*

**Sketch of proof.** The proof relies on similar concentration techniques than that of Proposition 3.1, while requiring the control of higher order deviation terms. Precise dependencies in $d, V, U, (M_{p,p'})_{p,p'}$ can be deduced from Equations (12) and (14) in the Appendix. □

Switching between the two parameterizations $(K, Q, V)$ and $(U, V)$, we observe that, for $U = K^\top Q$,

$$\nabla_K T^{K,Q,V} = Q(\nabla_U T^{U,V})^\top,$$
$$\nabla_Q T^{K,Q,V} = K \nabla_U T^{U,V},$$
$$\nabla_V T^{K,Q,V} = \nabla_V T^{U,V}.$$

Hence, the concentration results established for $\nabla_U$ and $\nabla_V$ in Proposition 3.4 can be directly transferred to obtain analogous bounds with respect to the original parameters $K, Q, V$.

If the measure perspective on transformers offers an elegant framework for handling i.i.d. input tokens, we see through these concentration bounds that this formalism can be leveraged for describing the non-asymptotic prompt setting ($L < +\infty$), which is often less tractable. Yet, one might still question the realism of the i.i.d. assumption—an issue we address in the following sections, where we show that it offers a meaningful lens through which to study the challenges of in-context learning.

# 4. General In-Context Learning

In-context learning consists in predicting the output associated with a new query input, given a prompt composed of input–output pairs. What distinguishes in-context learning is that each prompt implicitly defines its own task-specific relationship between inputs and outputs. Mathematically, it corresponds to minimizing (w.r.t. $U$ and $V$) a loss of the form

$$\mathcal{R}_L^{\text{ICL}}(U, V) = \mathbb{E}_{\mu \sim \mathcal{D}} \left[ \mathbb{E}_{(z_1,\dots,z_L) \sim \mu^{\otimes L}} [\mathcal{L}_\mu(T^{U,V}[\hat{\mu}_L])] \right],$$
$$\tag{2}$$

where $L$ is the prompt length and $\hat{\mu}_L = \frac{1}{L} \sum_{\ell=1}^L \delta_{z_\ell}$. Note that $z_\ell$ here denotes an *input/output pair*, so that with standard notations, we would have $z_\ell = (x_\ell, y_\ell) \in \mathbb{R}^{d+k}$, where $k$ would denote the label dimension. The task distribution $\mathcal{D}$ here denotes a distribution over the set of distributions so that $\mu$ is drawn according to $\mathcal{D}$ for each new task (i.e., prompt). In general, $\mathcal{L}_\mu$ quantifies the error of a prediction function $f$ on a new input and is thus of the form:

$$\mathcal{L}_\mu(f) = \mathbb{E}_{(x_q, y_q) \sim \mu} \left[ \ell \left( f \left( (x_q, 0)^\top \right), y_q \right) \right], \quad (3)$$

where in standard univariate ($k = 1$) regression settings, one can choose

$$\ell(\hat{z}, y) = \frac{1}{2}(\hat{z}_{d+1} - y)^2,$$

i.e., the squared error between the last coordinate of the output $\hat{z}$ and the true label $y$.

**Remark 4.1** (Distribution shift)**.** *Note that we could consider the case where $(x_q, y_q)$ differs in distribution from the*

*prompt variables. Such a mismatch would introduce an additional bias term in the generalization error. However, this effect is not measured here, as we are primarily concerned with the training phase—specifically, with the optimization of the attention-layer weights.*

In Section 3, we have established that, as the prompt length $L$ tends to infinity, both the attention output and its gradient converge to their counterparts obtained by taking the full measure $\mu$ as input. As a consequence, the in-context learning optimization process with finite, but sufficiently large, prompts is expected to closely mirror the behavior of the infinite-prompt regime, which directly targets the minimization of the infinite-length analogue of the risk $\mathcal{R}_L^{\text{ICL}}$ given by

$$\mathcal{R}_\infty^{\text{ICL}}(U, V) = \mathbb{E}_{\mu \sim \mathcal{D}} \left[ \mathcal{L}_\mu(T^{U,V}[\mu]) \right]. \quad (4)$$

In general, this optimization problem is considerably easier to analyze. For instance, when $\mu$ is a centered Gaussian measure, the resulting parametrization becomes linear in both $U$ and $V$ (see Lemma 2.1). Building on this observation, Theorem 4.3 below provides a general framework for comparing the optimization processes defined by the corresponding gradient flows

$$(\dot{U}_L(t), \dot{V}_L(t)) = -\nabla \mathcal{R}_L^{\text{ICL}}(U_L(t), V_L(t))$$
$$\text{and} \quad (U_L(0), V_L(0)) = (U_0, V_0), \tag{5}$$
$$(\dot{U}_\infty(t), \dot{V}_\infty(t)) = -\nabla \mathcal{R}_\infty^{\text{ICL}}(U_\infty(t), V_\infty(t))$$
$$\text{and} \quad (U_\infty(0), V_\infty(0)) = (U_0, V_0). \tag{6}$$

**Assumption 4.2.** *Assume that the following holds for some $\rho' > 0$:*

*(i) conditionally on $\mu$, $x_q$ is 1 sub-Gaussian and $(x_k, y_k) \sim \mu$ is $\sigma_\mu$ sub-Gaussian, with $\sigma_\mu \geq 1$, $\mathcal{D}$-almost surely.*

*(ii) $\mathbb{E}[\sigma_\mu^{12}] < \infty$ and for any $c > 0$, $\mathbb{E}_{\mu \sim \mathcal{D}} \left[ \frac{\ln(L)^4}{L^{c/\sigma_\mu^2}} \right] \xrightarrow[L \to \infty]{} 0$.*

*(iii) For all $(U, V) \in B_{2\rho'}$ and $p, p' \in \{0, 4, 8\}$, there exist $M_{p,p'}$ such that $\mathcal{D}$-almost surely*

$$\mathbb{E}_{(x_q, y_q) \sim \mu} \left[ \frac{\mathbb{E}_{z_1 \sim \mu}[\exp(z_1^\top U(x_q, 0)^\top) \|z_1\|^p \|x_q\|^{p'}]}{\mathbb{E}_{z_1 \sim \mu}[\exp(z_1^\top U(x_q, 0)^\top)]} \right]$$
$$\leq \sigma_\mu^{2p} M_{p,p'};$$

*where $B_{2\rho'}$ denotes the centered ball of radius $2\rho'$.*

Assumption 4.2 is fairly mild and ensures that the concentration results of Section 3 apply under well-behaved conditions. More specifically, item (i) assumes that the tokens are sub-Gaussian, a standard and convenient hypothesis in this line of work. Item (ii) further requires the (random)

sub-Gaussian parameter $\sigma_\mu$ itself to be light-tailed, ensuring that the resulting concentration behavior remains well controlled. Finally, item (iii) guarantees that the outputs of the attention layer possess sufficiently bounded moments.

**Theorem 4.3.** *Assume that $\ell$ is 1-smooth, $\nabla_1\ell(0,0) = 0$, $\mathcal{R}_\infty^{\mathrm{ICL}}$ is $C^2$. Assume in addition that the gradient flow $\begin{pmatrix} U_\infty(t) \\ V_\infty(t) \end{pmatrix}$ is contained within the centered ball $B_\rho$ of radius $\rho$ for some $\rho > 0$, and that Assumption 4.2 holds with $\rho' = \rho$.*

*Then for any $\varepsilon > 0$, there exists $L(\varepsilon)$ such that for any $L \geq L(\varepsilon)$, the solutions of the differential equations (5) and (6) satisfy*

$$\lim_{t \to \infty} \mathcal{R}_L^{\mathrm{ICL}}(U_L(t), V_L(t)) \leq \lim_{t \to \infty} \mathcal{R}_\infty^{\mathrm{ICL}}(U_\infty(t), V_\infty(t)) + \varepsilon.$$

**Sketch of proof.** The proof of Theorem 4.3 builds on Propositions 3.1 and 3.4, applied within the in-context learning framework. Specifically, as the prompt length $L$ increases, the finite-prompt risk $\mathcal{R}_L^{\mathrm{ICL}}$ converges to its infinite-prompt counterpart $\mathcal{R}_\infty^{\mathrm{ICL}}$, and the same convergence holds for their gradients. Leveraging these results, Grönwall's inequality yields a bound on the deviation between the finite-$L$ training trajectory and its infinite-prompt limit over any finite time horizon $t$.

By choosing $t$ sufficiently large, the limiting pair $(U_\infty(t), V_\infty(t))$ can be made arbitrarily close to its asymptotic convergence point. As a consequence, the corresponding finite-$L$ pair $(U_L(t), V_L(t))$ achieves a comparably low risk. The final limit statement then follows from the monotonic decrease of the risk throughout training. $\square$

This upper bound is particularly informative in regimes where the infinite-prompt model is easier to analyze or is known to perform well (most notably in the case of in-context linear regression, studied in the next section).

Theorem 4.3 states for a softmax attention layer that under mild regularity assumptions on the ICL loss, the parameters obtained after training over large, finite-length prompts obtain a population risk that is at least comparable (if not better) to the risk it would have obtained by training over infinite-length prompts. Although it provides guarantees only in terms of risk, its proof also yields (weaker) control over the training iterates themselves. In particular, Lemma E.1 implies that for any $\varepsilon > 0$, there exist sufficiently large values of $L$ and $T$ (with $T$ depending on $L$) such that

$$\|U_L(T) - \lim_{t \to \infty} U_\infty(t)\| + \|V_L(T) - \lim_{t \to \infty} V_\infty(t)\| \leq \varepsilon.$$

However, this result does not guarantee convergence of $(U_L(t), V_L(t))$ as $t \to \infty$, for a *fixed* prompt length $L$.

Rather, it ensures that

$$\lim_{L \to +\infty} \big(U_L(T(L)), V_L(T(L))\big) = \lim_{t \to \infty} \big(U_\infty(t), V_\infty(t)\big),$$

for an appropriately chosen sequence $T(L)$.

# 5. In-Context Linear Regression

In this section, we consider the problem of linear in-context learning, similar to that considered in (Zhang et al., 2024). In this case, each prompt corresponds to a specific linear regression task, with its own set of regression coefficients. More formally, here, each training prompt is associated with a linear regression task of parameter $w \in \mathbb{R}^d$ drawn at random according to $w \sim \mathcal{N}(0, \mathrm{I}_d)$. Consequently, each prompt is given by $[z_1, \ldots, z_L]$, where for any $\ell \leq L$, $z_\ell = (x_\ell, y_\ell)^\top \in \mathbb{R}^{d+1}$ are i.i.d. random variables such that

$$\begin{cases} w & \sim & \mathcal{N}(0, \mathrm{I}_d) \\ x_\ell & \sim & \mathcal{N}(0, \Sigma) \\ y_\ell & = & w^\top x_\ell. \end{cases} \tag{7}$$

The goal of the attention layer is then to predict the label $y_{\mathrm{q}}$, based on the (masked) query token $(x_{\mathrm{q}}, 0)^\top$. Using the notations of Section 4, this amounts to study in-context learning for univariate regression with

$$\begin{cases} w \sim \mathcal{N}(0, \mathrm{I}_d) \\ \mu = \mathcal{N}(0, \Gamma_w), \end{cases} \tag{8}$$

where $\Gamma_w = \begin{pmatrix} \Sigma & \Sigma w \\ (\Sigma w)^\top & \|w\|_\Sigma^2 \end{pmatrix}$. For this linear in-context task, considering an attention layer parameterized by $U, V \in \mathbb{R}^{(d+1) \times (d+1)}$, the theoretical risk for prompts of infinite length is given by

$$\begin{aligned} \mathcal{R}_\infty^{\mathrm{ICL}}(U, V) &= \mathbb{E}_{\mu \sim \mathcal{D}} \big[ \mathcal{L}_\mu(T^{U,V}[\mu]) \big] \\ &= \frac{1}{2} \mathbb{E}_{w \sim \mathcal{N}(0, \mathrm{I}_d)} \Big[ \mathbb{E}_{x_{\mathrm{q}} \sim \mathcal{N}(0, \Sigma)} \Big[ \\ &\quad \big( v_{d+1}^\top \Gamma_w U(x_{\mathrm{q}}, 0)^\top - w^\top x_{\mathrm{q}} \big)^2 \Big] \Big] \end{aligned}$$

where $v_{d+1}$ is the $(d + 1)$-th row of $V$. As observed by Zhang et al. (2024), note that the prediction concerns only the last coordinate of the transformer output. Consequently, the parameter $V$ affects the learning task solely through its last row, so its remaining components remain fixed during training.

When using a softmax architecture with a prompt of finite

length $L$, the theoretical risk reads as

$$\mathcal{R}_L^{\mathrm{ICL}}(U, V) = \mathbb{E}_{\mu \sim \mathcal{D}} \mathbb{E}_{(z_1, \ldots, z_L) \sim \mu^{\otimes L}} \left[ \mathcal{L}_\mu(T^{U,V}[\hat{\mu}_L]) \right]$$

$$= \frac{1}{2} \mathbb{E}_{w \sim \mathcal{N}(0, I_d)} \mathbb{E}_{\substack{(z_1, \ldots, z_L) \sim \mathcal{N}(0, \Gamma_w)^{\otimes L} \\ x_{\mathrm{q}} \sim \mathcal{N}(0, \Sigma)}} \left[ \left( \frac{\sum_{k=1}^L \exp\left(z_k^\top U(x_{\mathrm{q}}, 0)^\top\right) v_{d+1}^\top z_k}{\sum_{k=1}^L \exp\left(z_k^\top U(x_{\mathrm{q}}, 0)^\top\right)} - w^\top x_{\mathrm{q}} \right)^2 \right]$$

In what follows, we study the training dynamics of an attention layer with a softmax activation and a finite prompt, via the gradient flow associated with the risk $\mathcal{R}_L^{\mathrm{ICL}}$. Using previous results, this can be compared to the gradient flow dynamics with an infinite length prompt and a linear attention layer. To this end, we consider the same initialization scheme as in (Zhang et al., 2024).

**Assumption 5.1.** *The parameters $U, V$ are initialized by the following block matrix structure*

$$\begin{cases} U(0) & = \alpha \begin{pmatrix} \Theta\Theta^\top & \mathbf{0}_d \\ \mathbf{0}_d^\top & 0 \end{pmatrix} \\ V(0) & = \alpha \begin{pmatrix} \mathbf{0}_{d \times d} & \mathbf{0}_d \\ \mathbf{0}_d^\top & 1 \end{pmatrix}, \end{cases}$$

*where $\alpha \in \mathbb{R}$ and $\Theta \in \mathbb{R}^{d \times d}$ is any matrix satisfying both $\|\Theta\Theta^\top\|_F = 1$ and $\Theta\Sigma \neq \mathbf{0}_{d \times d}$.*

This block structure is preserved through training (as already noticed and used by Zhang et al., 2024, with linear attention), and allows a tractable analysis of training dynamics when $\alpha$ is chosen small enough. This assumption is mainly technical and does not appear to be necessary in practice, as illustrated by the experiments in Section 6.

**Theorem 5.2.** *Consider the in-context linear model described by* (7) *and an initialization of parameters given by Assumption 5.1 with $\alpha \in \left(0, \frac{\sqrt{2}}{d^{1/4} \|\Sigma\|_{\mathrm{op}}}\right)$. If $\Sigma$ is invertible, then for any $\varepsilon > 0$, there exists $L(\varepsilon)$ such that for any $L \geq L(\varepsilon)$, the attention parameters trained via gradient flow* (5) *satisfy*

$$\lim_{t \to \infty} \mathcal{R}_L^{\mathrm{ICL}}(U_L(t), V_L(t)) \leq \mathcal{R}^{\mathrm{ICL}, \star} + \varepsilon,$$

*where $\mathcal{R}^{\mathrm{ICL}, \star} = \lim_{t \to \infty} \mathcal{R}_\infty^{\mathrm{ICL}}(U_\infty(t), V_\infty(t))$ corresponds to the Bayes risk for the in-context linear regression task, which is attained at convergence by the gradient flow* (6) *for the infinite prompt setting.*

**Sketch of Proof.** The proof relies on applying Theorem 4.3 to the specific setting of linear in-context learning, which yields that

$$\lim_{t \to \infty} \mathcal{R}_L^{\mathrm{ICL}}(U_L(t), V_L(t)) \leq \lim_{t \to \infty} \mathcal{R}_\infty^{\mathrm{ICL}}(U_\infty(t), V_\infty(t)) + \varepsilon.$$

We then build on the results of Zhang et al. (2024), with minor adaptations to our framework, to establish the Bayes optimality of $(U_\infty, V_\infty)$. While the results of Zhang et al. (2024) are stated for a finite prompt length with (normalized) linear attention, both their analysis and conclusions extend seamlessly to the infinite-prompt regime ($L = \infty$). In particular, Equation (5.4) in Zhang et al. (2024), which governs the gradient-flow dynamics, remains valid in this limit, and the remainder of the argument carries over verbatim, yielding convergence of the gradient flow for infinite prompt and linear attention.

In this asymptotic regime, the softmax attention layer acts effectively as a linear operator on both the query token and prompt measure (see Lemma 2.1). As a result, the infinite-prompt softmax setting becomes analytically and conceptually equivalent to the corresponding infinite-prompt linear formulation, aligning precisely with the limiting case studied by Zhang et al. (2024). $\qquad\square$

**Remark 5.3.** *Zhang et al. (2024) even give a more precise result, characterizing the limits of $U_\infty$ and $V_\infty$ as*

$$U_\infty(t) \xrightarrow[t \to \infty]{} \mathrm{tr}(\Sigma^{-2})^{-1/4} \begin{pmatrix} \Sigma^{-1} & \mathbf{0}_d \\ \mathbf{0}_d^\top & 0 \end{pmatrix},$$

$$V_\infty(t) \xrightarrow[t \to \infty]{} \mathrm{tr}(\Sigma^{-2})^{1/4} \begin{pmatrix} \mathbf{0}_{d \times d} & \mathbf{0}_d \\ \mathbf{0}_d^\top & 1 \end{pmatrix}.$$

*These limit matrices are Bayes optimal for in-context linear regression (Zhang et al., 2024, Theorem 4.2), achieving a risk equal to 0, i.e., $\mathcal{R}^{\mathrm{ICL}, \star} = 0$, since there is no label noise here. Note that our results can also be extended to the presence of label noise, which would only add a variance term to the right-bottom corner entry of $\Gamma_w$, but we prefer to stick to Zhang et al. (2024) setting for clarity.*

*We also require $\Sigma$ to be invertible, inheriting the assumptions of Zhang et al. (2024), but their result (and ours henceforth) can be extended to a degenerate covariance matrix. This requires a slightly more careful analysis, where the limit parameters are given as a function of the pseudo-inverse $\Sigma^\dagger$. We refer to Appendix F.3 for more details about such an extension.*

Importantly, we do not provide explicit convergence rates here, as our objective is not to obtain sharp quantitative bounds, but rather to characterize the limiting behavior of the dynamics as $L$ becomes large. Deriving precise rates generally requires additional structural assumptions that are specific to the problem under consideration (for instance, a local Polyak–Łojasiewicz condition), together with technically involved arguments that do not directly rely on the infinite-prompt perspective.

Using such problem-specific techniques, Chen et al. (2024, Theorem 3.3) established precise convergence rates for in-context linear regression with softmax attention. Their

analysis moreover extends to multi-head and multitask settings, and accommodates distinct parametrizations for the key and query matrices. Their approach does not rely on comparisons with linear-attention architectures, but instead involves highly technical arguments tailored to the softmax setting. However, their guarantees hold only under isotropic covariates, i.e., when $\Sigma = I_d$. This isotropy assumption is crucial to their analysis, as it preserves a diagonal structure in the model parameters throughout training (see also He et al., 2025). By contrast, our proof strategy builds on the linear-attention analysis of Zhang et al. (2024), which relies on comparatively simpler mathematical arguments and remains valid beyond the isotropic-covariate regime.

# 6. Numerical experiments

This section presents numerical experiments illustrating our theoretical results. The experiments are implemented using a modified version of the codebase introduced by He et al. (2025), originally developed for a mechanistic interpretability study of in-context linear regression (as described in Section 5).

We train single-layer attention architectures using the standard parameterization $\theta = (O, V, K, Q)$ respectively denoting the output/value/key/query matrices[1]. Optimization is performed via stochastic gradient descent (SGD) using independently new mini-batches at each iteration.

The data are generated according to the model described in Equation (7), with dimension $d = 5$ and isotropic covariates $\Sigma = I_d$. Following the setup of He et al. (2025), the regression parameter is drawn as $w \sim \mathcal{N}(0, d^{-1}I_d)$, and additive label noise with variance $0.1$ is included. Complete experimental details are provided in Appendix A.

We first investigate the concentration behavior of softmax attention. Figure 1 illustrates the convergence of softmax attention—and its gradients—towards its normalized linear counterpart as the prompt length $L$ increases. In particular, the figure compares the outputs of the two models evaluated at identical parameter values, corresponding to the random initialization of the training dynamics, as well as their gradients with respect to the ICL risk. In agreement with our theoretical analysis, both the model outputs and the corresponding gradients become increasingly close as $L$ grows, eventually coinciding in the large-prompt limit.

Figure 2 reports the ICL risk at convergence for different prompt lengths $L$. In addition to single-layer attention models using either softmax or (normalized) linear

---

[1]The output matrix $O$ post-multiplies the model output, yielding the prediction $OT^{K,Q,V}[\hat{\mu}_L](z)$ for a prompt with empirical measure $\hat{\mu}_L$ and a query token $z$.

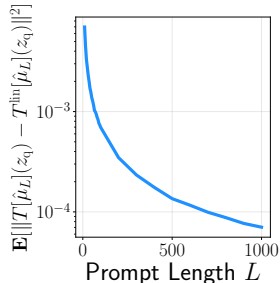
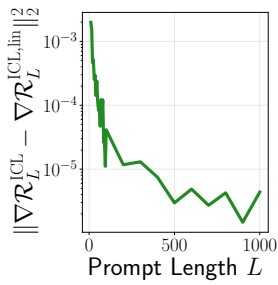

*(a)* Comparison of linear and softmax model outputs as $L$ increases.

*(b)* Comparison of linear and softmax model gradients (for ICL risk) as $L$ increases.

*Figure 1.* Concentration of output and gradient of softmax attention layer towards its linear counterpart, on Gaussian inputs.

attention, we compare their performance to the $L$-Bayes-optimal estimator, that is, the optimal estimator based on the $L$ observed prompt tokens (which corresponds to a well-tuned ridge regression; see, e.g., Williams and Rasmussen, 2006, Chapter 2). We observe that linear attention consistently outperforms softmax attention, although the performance gap narrows as $L$ increases. Moreover, the risks achieved by both models appear to converge to the label-noise level as $L \to \infty$. These findings are fully consistent with our theoretical analysis, and in particular with Theorem 5.2, which predicts convergence of both softmax and linear attention models to the Bayes-optimal predictor in the large-prompt limit. Finally, we note that the label noise is non-zero in this experiment, further confirming that our results remain valid in the presence of label noise, as discussed in Remark 5.3.

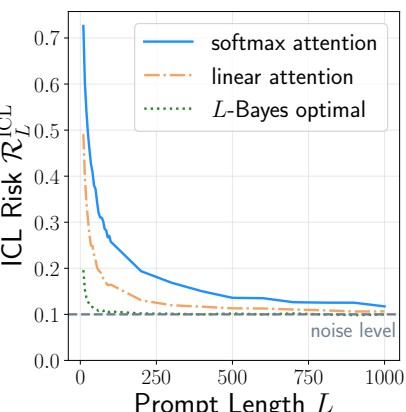

*Figure 2.* ICL risk of model at convergence. $L$ here refers to the prompt length used for both model training and the evaluation shown in the figure.

Figure 3 illustrates more precisely the similarity between the training dynamics of softmax and normalized linear attention, which we recall are theoretically equivalent in

the limit $L \to \infty$ under Gaussian inputs. Specifically, Figure 3 displays, for several values of $L$, the evolution during training of the squared norm of the difference between the parameters of the linear attention model and those of the softmax model. To enable a meaningful comparison of the two training trajectories, we use the same random seed for both parameter initialization and data generation at each training step. In agreement with our theoretical predictions, the distance between the two trajectories decreases as $L$ increases, indicating that the dynamics of softmax attention progressively align with those of normalized linear attention in the large-prompt regime.

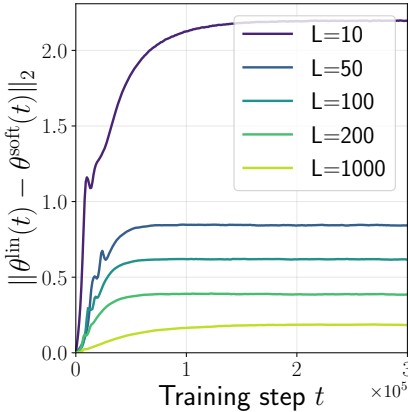

*Figure 3.* Evolution of distance between trained parameters obtained with softmax and linear attention for different prompt lengths.

Additional experiments with anisotropic Gaussian data are presented in Appendix A and lead to similar observations.

## 7. Conclusion

We developed a measure-based framework for softmax attention that characterizes its behavior in the infinite-prompt limit and provides non-asymptotic guarantees for finite prompts. This framework yields a faithful surrogate for analyzing training dynamics and shows that softmax and linear attention become increasingly similar in long-prompt regimes, with centered Gaussian inputs. As an application, we establish the first convergence guarantees for in-context linear regression with softmax attention under anisotropic Gaussian covariates. More broadly, our framework provides theoretical justification for analyzing simplified attention mechanisms, such as linear attention, without losing relevance to practical regimes encountered in transformer models. We believe that these tools can facilitate the study of training dynamics beyond Gaussian inputs and with more complex learning tasks, including, for instance, $n$-gram in-context learning (Edelman et al., 2024; Varre et al., 2025).

## Acknowledgments

The authors would like to extend special thanks to Cyril Letrouit for insightful discussions.

## Impact Statement

This paper presents work whose goal is to advance the field of Machine Learning. There are many potential societal consequences of our work, none which we feel must be specifically highlighted here.

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

# Appendix

## Table of Contents

## A. Additional experiments and experimental details

### A.1. Experimental details

The code used to run the experiments is based on the implementation of He et al. (2025) and is publicly available at github.com/eboursier/softmax_as_linear. The data for the ICL task are generated according to the following process:

$$\begin{cases} w & \sim & \mathcal{N}(0, d^{-1}\mathrm{I}_d) \\ x_\ell & \sim & \mathcal{N}(0, \Sigma) \\ y_\ell & = & w^\top x_\ell + \eta_\ell \quad \text{where} \quad \eta_\ell \sim \mathcal{N}(0, 0.1) \end{cases}$$

which matches the data-generating distribution used in the experiments of He et al. (2025). In Section 6, we focus on the isotropic setting $\Sigma = \mathrm{I}_d$ with $d = 5$. Training is performed using stochastic gradient descent with a learning rate of $10^{-3}$ for 500,000 steps. At each iteration, we sample a batch of $n_{\mathrm{batch}} = 128$ prompts, each of length $L$.

The softmax attention architecture is parameterized by matrices $O, V, K, Q$. Given a prompt $(z_1, \ldots, z_L)$ and a query token $z_{\mathrm{q}}$, the model outputs

$$T_{\mathrm{soft}}^{O,V,K,Q}[\hat{\mu}_L](z_{\mathrm{q}}) = O \frac{\sum_{\ell=1}^{L} \exp(\langle Q z_{\mathrm{q}}, K z_\ell \rangle) V z_\ell}{\sum_{\ell=1}^{L} \exp(\langle Q z_{\mathrm{q}}, K z_\ell \rangle)}.$$

The linear attention architecture uses the same parameterization $(O, V, K, Q)$ and produces

$$T_{\text{lin}}^{O,V,K,Q}[\hat{\mu}_L](z_{\text{q}}) = O\frac{1}{L}\sum_{\ell=1}^{L}\langle Qz_{\text{q}}, Kz_\ell\rangle Vz_\ell.$$

The additional normalization factor $1/L$, while not standard, ensures that the softmax and linear attention architectures exhibit comparable behavior in the limit $L \to \infty$ in our setting. The weights $O, V, K, Q$ are initialized at random following the standard Pytorch initialization.

To ensure a fair comparison of the training dynamics the random seed is fixed, so that both architectures are initialized with identical weights and the same prompt batches are presented at each training iteration.

All experiments were run on a personal laptop. Each run (corresponding to a fixed architecture and prompt length $L$) required approximately 30 minutes.

### A.2. Additional experiments

We provide additional experimental results in the appendix. Appendix A.2.1 reports the training dynamics for the experiments discussed in Section 6, while Appendix A.2.2 presents analogous experiments with anisotropic Gaussian covariates.

An observation not discussed in Section 6 is that training softmax attention architectures appears to be more stable than training linear attention architectures. In particular, for the linear models, optimization occasionally failed for some random initializations, requiring reruns with different random seeds to obtain successful training.

#### A.2.1. EVOLUTION OF ICL RISK OVER TRAINING

Figure 4 below illusrates the evolution of the ICL risk during training for $L \in \{10, 100, 1000\}$ and both softmax and linear activations.

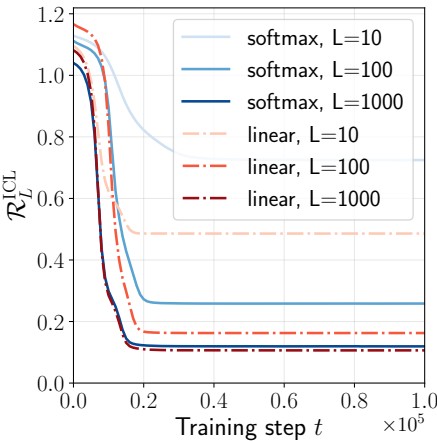

*Figure 4.* Evolution of ICL risk during training, for different activations and prompt lengths on isotropic data.

The evolution of the ICL risk is here rather natural—in contrast to the anisotropic case of Figure 8—and suggests that the convergence happens quite quickly, i.e., before 50 000 steps.

#### A.2.2. ANISOTROPIC GAUSSIAN COVARIATES

Since our work is the first to establish convergence guarantees for softmax attention in in-context linear regression with anisotropic Gaussian covariates, we include this section for completeness and provide experiments analogous to those in Section 6, but in the anisotropic setting.

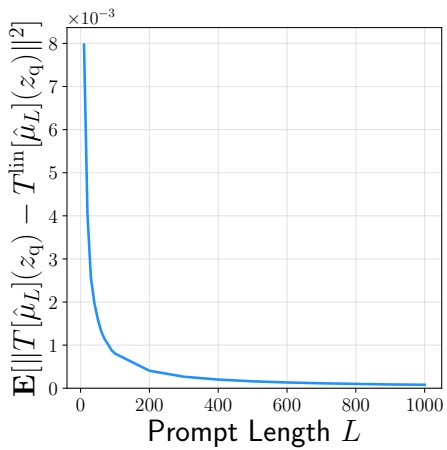
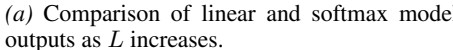
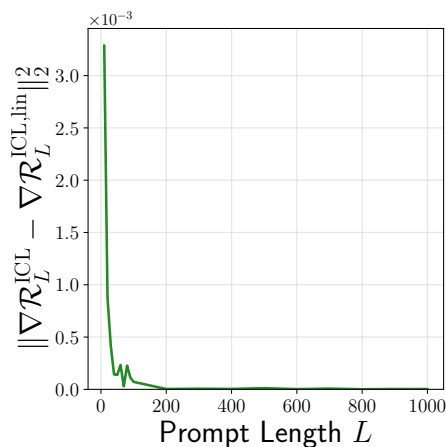

*(a)* Comparison of linear and softmax model outputs as $L$ increases.

*(b)* Comparison of linear and softmax model gradients (for ICL risk) as $L$ increases.

*Figure 5.* Concentration of output and gradient of softmax attention layer towards its linear counterpart, on anistropic Gaussian inputs.

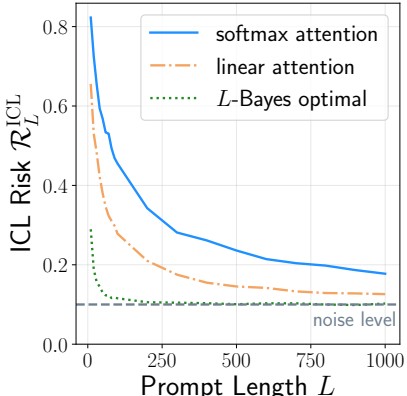

*Figure 6.* ICL risk of model at convergence for anistropic covariates. $L$ here refers to the prompt length used for both model training and the evaluation shown in the figure.

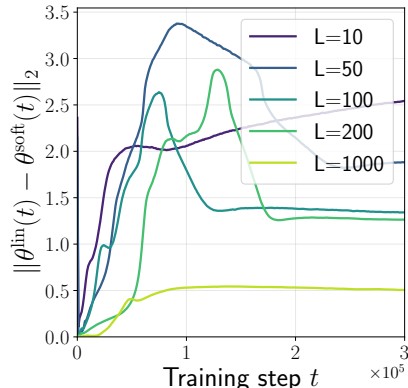

*Figure 7.* Evolution of distance between trained parameters obtained with softmax and linear attention for different prompt lengths for anisotropic covariates.

We consider the same experimental setup as described in Appendix A.1, with the sole modification that covariates are now sampled as $x_\ell \sim \mathcal{N}(0, \Sigma)$, where $\Sigma$ is a Kac–Murdock–Szegö matrix with parameter $\rho = 0.5$, i.e.,

$$\Sigma_{i,j} = 0.5^{|i-j|}.$$

In this representative anisotropic setting, we observe behavior qualitatively similar to the isotropic case: both softmax and linear attention models converge towards the Bayes-optimal predictor, and their training trajectories become increasingly close as $L \to \infty$.

In this typical anisotropic case, we get similar observations as in the isotropic case: both softmax and linear attention models converge towards the Bayes optimum, while their training trajectories are getting closer as $L \to \infty$.

However, in contrast to the isotropic case, several notable differences arise:

- The ICL risk converges more slowly towards the Bayes optimum as $L \to \infty$, as illustrated in Figure 6.

- The evolution of the difference between the two training trajectories is more erratic—and potentially non-monotonic—particularly for small values of $L$, as shown in Figure 7. Nevertheless, we still observe an overall

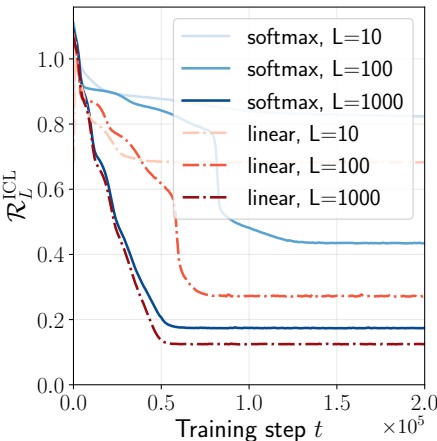

*Figure 8.* Evolution of ICL risk during training, for different activations and prompt lengths, on anisotropic data.

tendency for the trajectories to get closer as $L \to \infty$. For large prompt lengths (e.g., $L = 1000$), the evolution of this difference appears more regular, resembling the behavior observed in the isotropic case (Figure 3).

- For a fixed value of $L$, the training dynamics of the ICL risk are themselves more irregular compared to the isotropic setting, as illustrated in Figure 8.

Overall, these observations confirm our theoretical results, while also highlighting that the optimization process becomes more challenging in the presence of anisotropic covariates. In particular, larger prompt lengths may be required to achieve performance comparable to that obtained in the isotropic case.

## B. Results of Section 2

### B.1. Proof of Lemma 2.1

*Proof.* We define for any $z$ the skewed distribution $\mu_z$ by:

$$\mathrm{d}\mu_z(z') = \frac{\exp\big(\langle Qz, Kz'\rangle\big)}{\mathbb{E}_{z_0 \sim \mu}\Big[\exp\big(\langle Qz, Kz_0\rangle\big)\Big]}\mathrm{d}\mu(z')$$

Obviously, $\mu_z$ is a probability distribution. Moreover, note that, assuming $\Gamma$ is invertible,

$$\frac{\mathrm{d}\mu_z(z')}{\mathrm{d}z'} \propto \exp\Big(z'^\top K^\top Qz - \frac{1}{2}(z'-m)^\top\Gamma^{-1}(z'-m)\Big)$$
$$\propto \exp\Big(-\frac{1}{2}(z'-m-\Gamma K^\top Qz)^\top\Gamma^{-1}(z'-m-\Gamma K^\top Qz)\Big).$$

In particular, $\mu_z = \mathcal{N}(m + \Gamma K^\top Qz, \Gamma)$. Using this relation, we can now compute $T^{K,Q,V}[\mu](z)$:

$$T^{K,Q,V}[\mu](z) = \frac{V\int_{\mathbb{R}^d}\exp\Big(\langle Qz, Kz'\rangle\Big)z'\mathrm{d}\mu(z')}{\int_{\mathbb{R}^d}\exp\Big(\langle Qz, Kz'\rangle\Big)\mathrm{d}\mu(z')}$$
$$= V\mathbb{E}_{z' \sim \mu_z}[z']$$
$$= V(m + \Gamma K^\top Qz)$$

This equality still holds when $\Gamma$ is degenerate, by doing the same computation, replacing $\mathrm{d}z'$ by the Lebesgue measure on an appropriately chosen affine subspace of dimension $\mathrm{rank}(\Gamma)$.

Thus the output distribution is the pushforward through the affine map $z \mapsto Vm + V\Gamma K^\top Qz$ of the initial Gaussian density $\mathcal{N}(m, \Gamma)$. It is therefore also Gaussian, given by $\mathcal{N}(V(\mathrm{I}_d + \Gamma K^\top Q)m, (V\Gamma K^\top Q)\Gamma(V\Gamma K^\top Q)^\top)$. □

## B.2. Relation to linear self-attention

Normalized standard linear self-attention (LSA) is given by the parametrization:

$$T_{\text{lin}}^{V,K,Q}[\hat{\mu}_L](z_{\text{q}}) = \frac{1}{L} \sum_{\ell=1}^{L} \langle Q z_{\text{q}}, K z_{\ell} \rangle V z_{\ell}.$$

In contrast to the standard definition, we here include the normalization factor $\frac{1}{L}$ so that the limit as $L \to \infty$ is well-defined.

A straightforward application of the law of large numbers then yields the pointwise limit

$$T_{\text{lin}}^{V,K,Q}[\mu](z_{\text{q}}) = V \mathbb{E}_{z \sim \mu}[z z^\top] K^\top Q z_{\text{q}}.$$

This expression coincides with the softmax attention output given in Lemma 2.1 in the special case of *centered* Gaussian inputs. Accordingly, for centered Gaussian inputs, softmax and linear attention are equivalent.

# C. Proofs of Section 3: Concentration of Attention

The output $T^{K,Q,V}$ only depends on $K, Q$ by their product $K^\top Q$. In consequence, we use in the section the parametrization $U = K^\top Q$ and $T^{K,Q,V} = T^{U,V}$ for conciseness. Yet, since this section solely concerns concentration of the output of $T^{K,Q,V}$, our result hold both for $T^{U,V}$ and $T^{K,Q,V}$.

## C.1. Preliminaries for the high-probability bound (Propositons 3.1 and 3.4)

Lemma C.1 below allows to provide a high probability (over prompt tokens $z_\ell$) concentration bound for a fixed input token $z$.

**Lemma C.1.** *If $\mu$ is centered and $\sigma$ sub-Gaussian, then for any $\delta \geq \exp\left(-\frac{L}{8} \exp(-2\|Uz\|^2)\right)$ and $z \in \mathbb{R}^d$, with probability larger than $1 - 4\delta$,*

$$\left\| \frac{\sum_{k=1}^{L} \exp(z_k^\top U z) f(z_k)}{\sum_{k=1}^{L} \exp(z_k^\top U z)} - \frac{\mathbb{E}_{Z_1 \sim \mu}[\exp(Z_1^\top U z) f(Z_1)]}{\mathbb{E}_{Z_1 \sim \mu}[\exp(Z_1^\top U z)]} \right\|^2 \leq 27 \beta_{L,\delta}^2,$$

*where $\beta_{L,\delta} = \exp(2\sigma^2 \|Uz\|^2) \cdot \mathbb{E}_{Z_1}[\|f(Z_1)\|^4]^{1/4} \cdot \sqrt{\frac{1}{\delta L}}$.*

*Proof of Lemma C.1.* For a fixed $z$, we introduce the notation

$$\begin{cases} R_L & := \sum_{i=1}^{L} \exp(z_i^\top U z) f(z_i) \\ S_L & := \sum_{i=1}^{L} \exp(z_i^\top U z) \end{cases}$$

in which we omit the dependence in $z$. Then, by definition

$$\frac{\sum_{k=1}^{L} \exp(z_k^\top U z) f(z_k)}{\sum_{k=1}^{L} \exp(z_k^\top U z)} =: \frac{R_L}{S_L}. \tag{9}$$

To control the quantity in Lemma C.1, we proceed as follows

$$\left\| \frac{R_L}{S_L} - \frac{R}{S} \right\| = \left\| \frac{R_L}{S_L} - \frac{R}{S_L} + \frac{R}{S_L} - \frac{R}{S} \right\|$$

$$\leq \frac{1}{S_L} \|R_L - R\| + \|R\| \left| \frac{1}{S_L} - \frac{1}{S} \right|, \tag{10}$$

where $R = L \cdot \mathbb{E}_{Z_1}[\exp(Z_1^\top U z) f(Z_1)]$ and $S = L \cdot \mathbb{E}_{Z_1}[\exp(Z_1^\top U z)]$ are the expected values of $R_L$ and $S_L$.

**Control of $S_L$.** For a fixed $z \in \mathbb{R}^d$, set $M_\ell = \exp(z_\ell^\top U z)$, which is by sub-Gaussianity of $z_\ell$ such that

$$\mathbb{E}\left[M_\ell\right] = \frac{S}{L} \leq \exp(\sigma^2 \|Uz\|^2/2) \quad \text{and} \quad \mathbb{E}\left[M_\ell^2\right] \leq \exp(2\sigma^2\|Uz\|^2).$$

To first control a lower bound on $S_L$, we proceed as follows. By setting $S_L = \sum_{\ell=1}^L M_\ell$ and observing that $M_\ell$ are almost surely non-negative, Theorem 7 of Chung and Lu (2006) gives for any $\alpha_- \in (0,1)$

$$\mathbb{P}\left(S_L \leq (1-\alpha_-)S\right) \leq \exp\left(-\frac{\alpha_-^2 S^2}{2L\mathbb{E}[M_1^2]}\right)$$
$$\leq \exp\left(-\frac{\alpha_-^2 L \exp(-2\sigma^2\|Uz\|^2)}{2}\right),$$

where we used that $S = L \cdot \mathbb{E}[M_1] \geq L$ by Jensen's inequality. Letting $\alpha_- = \exp(\sigma^2\|Uz\|^2)\sqrt{\frac{2\ln(1/\delta)}{L}}$, this yields that with probability at least $\delta$,

$$S_L \geq (1-\alpha_-)S.$$

The concentration upper bound for $S_L$ is obtained by resorting to the heavy tail concentration bound of Bubeck et al. (2013, Lemma 3), which directly yields here that with probability larger than $1-\delta$,

$$S_L \leq S + \sqrt{\frac{3L\mathbb{E}[M_1^2]}{\delta}} \leq \left(1 + \exp\left(\sigma^2\|Uz\|^2\right)\sqrt{\frac{3}{\delta L}}\right)S.$$

Thus we have shown that with probability at least $1-2\delta$,

$$(1-\alpha_-)S \leq S_L \leq (1+\alpha_+)$$

where $\alpha_- = \exp(\sigma^2\|Uz\|^2)\sqrt{\frac{2\ln(1/\delta)}{L}}$ and $\alpha_+ = \exp\left(\sigma^2\|Uz\|^2\right)\sqrt{\frac{3}{\delta L}}$. This inequality directly implies the following inequalities

$$\frac{1}{1+\alpha_+} \cdot \frac{1}{S} \leq \frac{1}{S_L} \leq \frac{1}{1-\alpha_-} \cdot \frac{1}{S},$$
$$-\frac{\alpha_+}{1+\alpha_+} \cdot \frac{1}{S} \leq \frac{1}{S_L} - \frac{1}{S} \leq \frac{\alpha_-}{1-\alpha_-} \cdot \frac{1}{S}.$$

Finally, we thus get that, with probability at least $1-2\delta$,

$$\left|\frac{1}{S_L} - \frac{1}{S}\right| \leq \max\left(\alpha_+, \frac{\alpha_-}{1-\alpha_-}\right)\frac{1}{S}.$$

**Control of $R_L$.** Define $\tilde{Q}_i = \exp(z_i^\top U z)f(z_i)$. Then, using Cauchy-Schwarz inequality:

$$\mathbb{E}\left[\|\tilde{Q}_i\|^2\right] \leq \sqrt{\mathbb{E}\left[\exp(4z_i^\top U z)\right] \cdot \mathbb{E}\left[\|f(z_i)\|^4\right]}$$
$$\leq \exp(4\sigma^2\|Uz\|^2) \cdot \sqrt{\mathbb{E}[\|f(z_i)\|^4]}.$$

Lemma 3 of Bubeck et al. (2013) can directly be extended to random vectors to the following lemma.

**Lemma C.2.** *Let $Q_1, \ldots, Q_L$ be an i.i.d. sequence of random vectors in $\mathbb{R}^d$ such that $\mathbb{E}[\|Q_i\|^2] \leq u$. Then for any $\delta \in (0,1)$, with probability at least $1-2\delta$:*

$$\left\|\sum_{i=1}^L Q_i - \mathbb{E}[Q_i]\right\| \leq \sqrt{\frac{3uL}{\delta}}.$$

We can use again this heavy tail concentration bound so that with probability larger than $1 - 2\delta$,

$$\|R_L - R\| \leq \sqrt{3 \frac{L \exp(4\sigma^2 \|Uz\|^2) \sqrt{\mathbb{E}[\|f(z_1)\|^4]}}{\delta}}$$

$$= \exp(2\sigma^2 \|Uz\|^2) \cdot \mathbb{E}[\|f(z_1)\|^4]^{1/4} \cdot \sqrt{\frac{3L}{\delta}}.$$

**Gluing things together.** Finally using Equation (10) and the fact that $S \geq L$, it comes with probability larger than $1 - 4\delta$ that

$$\left\| \frac{R_L}{S_L} - \frac{R}{S} \right\| \leq \max\left(\alpha_+, \frac{\alpha_-}{1 - \alpha_-}\right) \left\| \frac{R}{S} \right\| + \frac{\beta}{1 - \alpha_-},$$

where $\alpha_+ = \exp\left(\sigma^2 \|Uz\|^2\right) \sqrt{\frac{3}{\delta L}}$, $\alpha_- = \min\left(\exp(\sigma^2 \|Uz\|^2)\sqrt{\frac{2 \ln(1/\delta)}{L}}, 1\right)$ and $\beta = \exp(2\sigma^2 \|Uz\|^2) \cdot$
$\mathbb{E}[\|f(z_1)\|^4]^{1/4} \cdot \sqrt{\frac{3}{\delta L}}$.

Note that

$$\alpha_- \leq \frac{1}{2} \iff \delta \geq \underbrace{\exp\left(-\frac{L}{8} \exp(-2\sigma^2 \|Uz\|^2)\right)}_{=: \delta_{\min}}$$

so that for $\delta \geq \delta_{\min}$, we can ensure that $1/(1 - \alpha_-) \leq 2$ and then, with probability larger than $1 - 4\delta$,

$$\left\| \frac{R_L}{S_L} - \frac{R}{S} \right\| \leq \max\left(\alpha_+, 2\alpha_-\right) \left\| \frac{R}{S} \right\| + 2\beta$$

$$\leq \left\| \frac{R}{S} \right\| \alpha_+ + 2\beta,$$

using that $\alpha_+ \geq 2\alpha_-$. Now note that

$$\left\| \frac{R}{S} \right\| = \left\| \frac{\mathbb{E}_{Z_1}[\exp(Z_1^\top Uz) f(Z_1)]}{\mathbb{E}_{Z_1}[\exp(Z_1^\top Uz)]} \right\| \leq \sqrt{\mathbb{E}_{Z_1}[\exp(2Z_1^\top Uz)] \cdot \mathbb{E}_{Z_1}[\|f(Z_1)\|^2]}$$

$$\leq \exp(\sigma^2 \|Uz\|^2) \sqrt{\mathbb{E}_{Z_1}[\|f(Z_1)\|^2]} \qquad (11)$$

Then, with probability at least $1 - 4\delta$, for $\delta \geq \delta_{\min}$

$$\left\| \frac{R_L}{S_L} - \frac{R}{S} \right\| \leq \exp(\sigma^2 \|Uz\|^2) \sqrt{\mathbb{E}[\|f(z_1)\|^2]} \cdot \exp\left(\sigma^2 \|Uz\|^2\right) \sqrt{\frac{3}{\delta L}}$$

$$+ 2 \exp(2\sigma^2 \|Uz\|^2) \cdot \mathbb{E}[\|f(z_1)\|^4]^{1/4} \cdot \sqrt{\frac{3}{\delta L}}$$

$$\leq 3 \exp(2\sigma^2 \|Uz\|^2) \mathbb{E}[\|f(z_1)\|^4]^{1/4} \sqrt{\frac{3}{\delta L}},$$

which yields Lemma C.1.

$\square$

**Corollary C.3.** *If $\mu$ is $\sigma$ sub-Gaussian, for any $z \in \mathbb{R}^d$ and $\delta \geq \delta_0 := \frac{64 \exp(4\sigma^2 \|Uz\|^2)}{L^2}$,*

$$\mathbb{P}\left(\left\| \frac{\sum_{k=1}^L \exp(z_k^\top Uz) f(z_k)}{\sum_{k=1}^L \exp(z_k^\top Uz)} - \frac{\mathbb{E}_{Z_1}[\exp(Z_1^\top Uz) f(Z_1)]}{\mathbb{E}_{Z_1}[\exp(Z_1^\top Uz)]} \right\|^2 \geq \frac{C}{\delta L}\right) \leq \delta,$$

*where $C = C(U, f, z) = 108 \exp(4\sigma^2 \|Uz\|^2) \sqrt{\mathbb{E}_{Z_1}[\|f(Z_1)\|^4]}$.*

*Proof.* Since $e^{-x} \leq \frac{1}{x^2}$ on $(0, \infty)$, we have $\delta_0 \geq \delta_{\min}$. Lemma C.1 then yields that with probability $1 - 4\delta$,

$$\left\| \frac{\sum_{k=1}^{L} \exp(z_k^\top U z) f(z_k)}{\sum_{k=1}^{L} \exp(z_k^\top U z)} - \frac{\mathbb{E}_{Z_1}[\exp(Z_1^\top U z) f(Z_1)]}{\mathbb{E}_{Z_1}[\exp(Z_1^\top U z)]} \right\|^2 \leq 27 \exp(4\sigma^2 \|U z\|^2) \sqrt{\mathbb{E}_{Z_1}[\|f(Z_1)\|^4]} \frac{1}{\delta L}.$$

Replacing $\delta$ by $\delta/4$ leads to the desired result: with probability larger than $1 - \delta$, it holds that

$$\left(T^{U,V}[\hat{\mu}_L](z) - T^{U,V}[\mu](z)\right)^2 \leq 108 \exp(4\sigma^2 \|U z\|^2) \sqrt{\mathbb{E}_{Z_1}[\|f(Z_1)\|^4]} \frac{1}{\delta L}.$$

$\square$

## C.2. Concentration of the second order over input tokens (fixed query)

Lemma C.4 below allows to provides an expected concentration bound for a fixed input token $z$. This bound does not directly derive from Corollary C.3, but also requires a careful bounding of the squared difference in the small probability event where the concentration of Corollary C.3 does not hold.

**Lemma C.4.** *Assume $\mu$ is centered, $\sigma$ sub-Gaussian and there is $C_f \in \mathbb{R}_+$ such that $\sup_{z \in \mathbb{R}^d} \frac{\|f(z)\|}{\|z\|^p} \leq C_f$ for some $p \in \{1, 2\}$. Then for any $z \in \mathbb{R}^d$,*

$$\mathbb{E}_{z_1,\dots,z_L \sim \mu^{\otimes L}} \left[ \left\| \frac{\sum_{k=1}^{L} \exp(z_k^\top U z) f(z_k)}{\sum_{k=1}^{L} \exp(z_k^\top U z)} - \frac{\mathbb{E}_{Z_1}[\exp(Z_1^\top U z) f(Z_1)]}{\mathbb{E}_{Z_1}[\exp(Z_1^\top U z)]} \right\|^2 \right] \leq \frac{2C \ln(L)}{L} +$$

$$2 \cdot 16^p \sigma^{2p} L p \, C_f e^{2d} \left( 1 + \mathbb{1}_{p=2} \frac{\sqrt{L}}{22\sigma^2} \cdot \mathbb{E}[\|Z_1\|^8]^{1/4} \right)$$

$$\cdot \exp\left( -\left( \frac{L^{1/p}}{2^{1/p}} - \exp(\frac{2}{p}\sigma^2 \|U z\|^2) \right) \cdot \frac{\mathbb{E}[\|f(Z_1)\|^4]^{1/2p}}{16\sigma^2 C_f^{2/p}} \right),$$

*where $C = C(U, f, z) = 108 \exp(4\sigma^2 \|U z\|^2) \sqrt{\mathbb{E}_{Z_1}[\|f(Z_1)\|^4]}$*

*Proof.* Using the same notations as in the proof of Lemma C.1, it comes for $C$ defined by Corollary C.3 and $t_0 = \frac{16C}{27\delta_0 L}$:

$$\mathbb{E}_{z_1,\dots z_L} \left[ \left\| \frac{R_L}{S_L} - \frac{R}{S} \right\|^2 \right] = \int_0^\infty \mathbb{P}\left[ \left\| \frac{R_L}{S_L} - \frac{R}{S} \right\|^2 \geq t \right] \mathrm{d}t$$

$$\leq \frac{C}{L} + \int_{\frac{C}{L}}^{t_0} \frac{C}{Lt} \mathrm{d}t + \int_{t_0}^\infty \mathbb{P}\left( \max_\ell \|f(z_\ell)\|^2 \geq \frac{t}{2} - \frac{R^2}{S^2} \right) \mathrm{d}t,$$

where we used Corollary C.3 for $t \leq t_0$, and for $t > t_0$ that

$$\left\| \frac{R_L}{S_L} - \frac{R}{S} \right\|^2 \geq t \qquad \implies \qquad 2 \left\| \frac{R_L}{S_L} \right\|^2 + 2 \left\| \frac{R}{S} \right\|^2 \leq t$$

$$\implies \qquad \left\| \frac{R_L}{S_L} \right\|^2 \geq \frac{t}{2} - \left\| \frac{R}{S} \right\|^2$$

$$\implies \qquad \max_\ell \|f(z_\ell)\|^2 \geq \frac{t}{2} - \left\| \frac{R}{S} \right\|^2.$$

The last line here used the fact that $\frac{R_L}{S_L}$ is in the convex hull of $(z_\ell)_\ell$ by definition of the softmax. Now using that

$\mathbb{P}(\max_\ell \|f(z_\ell)\|^2 \geq u) \leq L\mathbb{P}(\|f(z_1)\|^2 \geq u)$ by union bound,

$$
\begin{aligned}
\mathbb{E}_{z_1,\ldots z_L}\left[\left\|\frac{R_L}{S_L} - \frac{R}{S}\right\|^2\right] &\leq \frac{C}{L} + \frac{C}{L}\left(\ln(t_0) + \ln(L/C)\right) + 2L\int_{\left(\frac{t_0}{2} - \frac{\|R\|^2}{S^2}\right)_+}^{\infty} \mathbb{P}\left(C_f^2\|z_1\|^{2p} \geq t\right)\,\mathrm{d}t \\
&\leq \frac{C(1+\ln(t_0 L/C))}{L} + 2L\int_{\left(\frac{t_0}{2} - \frac{\|R\|^2}{S^2}\right)_+}^{\infty} \mathbb{P}\left(\|z_1\|^2 \geq \frac{t^{1/p}}{C_f^{2/p}}\right)\,\mathrm{d}t \\
&\leq \frac{C(1-\ln(\delta_0) + \ln(16/27))}{L} + 2Le^{2d}\int_{\left(\frac{t_0}{2} - \frac{\|R\|^2}{S^2}\right)_+}^{\infty} \exp\left(-\frac{t^{1/p}}{16\sigma^2 C_f^{2/p}}\right)\,\mathrm{d}t \\
&\leq \frac{C(1-\ln(\delta_0))}{L} + 2Lpe^{2d}\int_{\left(\frac{t_0}{2} - \frac{\|R\|^2}{S^2}\right)_+^{1/p}}^{\infty} u^{p-1}\exp\left(-\frac{u}{16\sigma^2 C_f^{2/p}}\right)\,\mathrm{d}u,
\end{aligned}
$$

where we use that for $\sigma$ sub-Gaussian random variables, for any $t$, $\mathbb{P}\left(\|z_1\|^2 \geq t\right) \leq e^{2d-t/16\sigma^2}$ (see Lemma D.2 in Appendix D). By definition of $\delta_0$,

$$
1 - \ln(\delta_0) = 1 + 2\ln(L) - 6\ln(2) - 4\sigma^2\|Uz\|^2 \leq 2\ln(L).
$$

Moreover, recall that

$$
t_0 = L\sqrt{\mathbb{E}[\|f(Z_1)\|^4]} \qquad \text{and} \qquad \frac{\|R\|^2}{S^2} \leq \exp(2\sigma^2\|Uz\|^2)\sqrt{\mathbb{E}[\|f(Z_1)\|^4]}.
$$

By definition of $C_f$, $\mathbb{E}[\|f(Z_1)\|^4] \leq C_f^4\mathbb{E}[\|Z_1\|^{4p}]$, so that

$$
\left(\frac{t_0}{2} - \frac{\|R\|^2}{S^2}\right)_+^{1/p} \cdot \frac{1}{16\sigma^2 C_f^{2/p}} \geq \left(\frac{L}{2} - \exp(2\sigma^2\|Uz\|^2)\right)_+^{1/p} \cdot \frac{\mathbb{E}[\|f(Z_1)\|^4]^{1/2p}}{16\sigma^2 C_f^{2/p}}
$$

$$
\text{and} \qquad \left(\frac{t_0}{2} - \frac{\|R\|^2}{S^2}\right)_+^{1/p} \cdot \frac{1}{16\sigma^2 C_f^{2/p}} \leq \frac{L^{1/p}}{2^{4+1/p}\sigma^2} \cdot \mathbb{E}[\|Z_1\|^{4p}]^{1/2p}.
$$

Moreover for the remaining integral, note that

$$
\begin{aligned}
\int_{\left(\frac{t_0}{2} - \frac{\|R\|^2}{S^2}\right)_+^{1/p}}^{\infty} u^{p-1}\exp\left(-\frac{u}{16\sigma^2 C_f^{2/p}}\right)\,\mathrm{d}u &= 16^p\sigma^{2p}C_f\int_{\left(\frac{t_0}{2} - \frac{\|R\|^2}{S^2}\right)_+^{1/p}/(16\sigma^2 C_f^{2/p})}^{\infty} v^{p-1}\exp\left(-v\right)\,\mathrm{d}v \\
&= 16^p\sigma^{2p}C_f\Gamma\left(p, \left(\frac{t_0}{2} - \frac{\|R\|^2}{S^2}\right)_+^{1/p}/(16\sigma^2 C_f^{2/p})\right),
\end{aligned}
$$

where $\Gamma(\cdot,\cdot)$ is the upper incomplete Gamma function. Since $p \in \{1,2\}$, we can use the upper bound $\Gamma(p,x) \leq (1 + x\mathbb{1}_{p=2})e^{-x}$ for all $x \geq 0$ (Pinelis, 2020) so that finally

$$
\begin{aligned}
\mathbb{E}_{z_1,\ldots z_L}\left[\left\|\frac{R_L}{S_L} - \frac{R}{S}\right\|^2\right] &\leq \frac{2C\ln(L)}{L} + 2\cdot 16^p\sigma^{2p}LpC_f e^{2d}\left(1 + \mathbb{1}_{p=2}\frac{\sqrt{L}}{22\sigma^2}\cdot\mathbb{E}[\|Z_1\|^8]^{1/4}\right) \\
&\quad \cdot \exp\left(-\left(\frac{L}{2} - \exp(2\sigma^2\|Uz\|^2)\right)_+^{1/p} \cdot \frac{\mathbb{E}[\|f(Z_1)\|^4]^{1/2p}}{16\sigma^2 C_f^{2/p}}\right) \\
&\leq \frac{2C\ln(L)}{L} + 2\cdot 16^p\sigma^{2p}LpC_f e^{2d}\left(1 + \mathbb{1}_{p=2}\frac{\sqrt{L}}{22\sigma^2}\cdot\mathbb{E}[\|Z_1\|^8]^{1/4}\right) \\
&\quad \cdot \exp\left(-\left(\frac{L^{1/p}}{2^{1/p}} - \exp(\frac{2}{p}\sigma^2\|Uz\|^2)\right) \cdot \frac{\mathbb{E}[\|f(Z_1)\|^4]^{1/2p}}{16\sigma^2 C_f^{2/p}}\right),
\end{aligned}
$$

which concludes the proof. $\qquad\square$

## C.3. Concentration of the second order over both query and input tokens

The goal of this section is now to prove an expected concentration bound for finite length prompt, when taking expectation **with respect to both prompt sequence and input token**. Again, we will have to resort to different bounds, depending on whether we are in high probability events where different vector norms and deviations are well controlled or not. First of all, we need moment conditions given by Assumption C.5 below.

**Assumption C.5.** *Assume that $f, g$ satisfy the following properties for the family of probability distributions $P \subseteq \mathcal{P}(\mathbb{R}^d)$ and the integer $p \in \mathbb{N}$:*

*(i) there is $C_f \in \mathbb{R}_+$ such that $\sup_{z \in \mathbb{R}^d} \frac{\|f(z_1)\|}{\|z_1\|^p} \leq C_f$;*

*(ii) for any $U \in \mathbb{R}^{d \times d}$ and $\mu \in P$ that is $\sigma$ sub-Gaussian, $\mathbb{E}_z \left[ \frac{\left\| \mathbb{E}_{Z_1} \left[ \exp(Z_1^\top U z) f(Z_1) g(z) \right] \right\|^4}{\mathbb{E}_{Z_1} \left[ \exp(Z_1^\top U z) \right]^4} \right] \leq \sigma^{8p} c(f, g, U) < +\infty$;*

*(iii) $\mathbb{E}_z[\|g(z)\|^4] < +\infty$.*

Note that we could use a more general Assumption C.5 (ii), where the scaling in $\sigma$ could be of arbitrary exponent, without any significant change in our following results. The scaling chosen here is just for the sake of presentation, as it corresponds to the one we will encounter in our in-context learning application. In addition, remark that $p$ characterizes the homogeneity degree of function $f$.

**Lemma C.6** (General finite-length bound). *Assume that $\mu$ and $\nu$ are both centered, respectively $\sigma$ and $1$ sub-Gaussian distributions with $\sigma \geq 1$. Moreover, consider functions $f, g$ satisfying Assumption C.5 for $p \in \{1, 2\}$ and $\mu \in P$, assuming that the product $f(z_1) g(z)$ is legit and that $\|f(z_1) g(z)\| \leq \|f(z_1)\| \|g(z)\|$. Then there exist constants $c_1, c_2 > 0$ depending solely on $d, f, g, p, P, U$ such that,*

$$\mathbb{E} \left[ \left\| \frac{\sum_{k=1}^L \exp(z_{\text{query}}^\top U^\top z_k) f(z_k) g(z)}{\sum_{k=1}^L \exp(z_{\text{query}}^\top U^\top z_k)} - \frac{\mathbb{E}_{Z_1}[\exp(z_{\text{query}}^\top U^\top Z_1) f(Z_1) g(z)]}{\mathbb{E}_{Z_1}[\exp(z_{\text{query}}^\top U^\top Z_1)]} \right\|_{L^2(\nu)}^2 \right] \leq c_1 \sigma^{6p} \frac{\ln(L)^p}{L^{c_2/\sigma^2}}$$

*where $Z_1 \sim \mu$ and the expectation is taken over $(z_1, \ldots, z_L) \sim \mu^{\otimes L}$.*

The proof provides a more explicit bound of the form

$$\mathbb{E} \left[ \left\| \frac{\sum_{k=1}^L \exp(z_{\text{query}}^\top U^\top z_k) f(z_k)}{\sum_{k=1}^L \exp(z_{\text{query}}^\top U^\top z_k)} - \frac{\mathbb{E}_{Z_1}[\exp(z_{\text{query}}^\top U^\top Z_1) f(Z_1)]}{\mathbb{E}_{Z_1}[\exp(z_{\text{query}}^\top U^\top Z_1)]} \right\|_{L^2(\nu)}^2 \right] \tag{12}$$

$$\leq \tilde{c}_1 \sigma^{2p} L^{3/2} \exp(-\tilde{c}_2 L^{1/p} / \sigma^2) + \tilde{c}_3 \frac{\ln(L)}{L^{1/3}} + \tilde{c}_4 \sigma^{4p} \frac{\ln(L)^p}{L^{1/(192\sigma^2 \|U\|_{\text{op}}^2)}},$$

where $\tilde{c}_1, \tilde{c}_2, \tilde{c}_3, \tilde{c}_4$ are positive constants depend solely on $d, f, g, p, P, U$.

Note that Lemma C.6 above provides a more general concentration than just on the transformer output, which would correspond to $f(z_k) = V z_k$ and $g(z) = 1$. This is because this more general form will also be used to control the concentration of gradients, via other choices of $f$ and $g$. Note that such a general bound could even be used to control higher order derivatives if desired.

*Proof.* Again in this section, we follow the notation used in the proofs of Lemmas C.1 and C.4. To get a bound on the

expectation, we split the computation as follows for some $B \in \mathbb{R}_+$

$$\mathbb{E}_z \left[ \mathbb{E}_{z_1,\ldots,z_L} \left[ \left\| \frac{R_L}{S_L} g(z) - \frac{R}{S} g(z) \right\|^2 \right] \right]$$

$$\leq \mathbb{E}_z \left[ \mathbb{E}_{z_1,\ldots,z_L} \left[ \left\| \frac{R_L}{S_L} - \frac{R}{S} \right\|^2 \right] \|g(z)\|^2 \mathbb{1}_{\|z\| \leq B} \right] + \mathbb{E}_z \left[ \mathbb{E}_{z_1,\ldots,z_L} \left[ \left\| \frac{R_L}{S_L} g(z) - \frac{R}{S} g(z) \right\|^2 \right] \mathbb{1}_{\|z\| > B} \right]$$

$$\leq \mathbb{E}_z \left[ \frac{2C(z)\ln(L)}{L} \|g(z)\|^2 \mathbb{1}_{\|z\| \leq B} \right] + \mathbb{E}_z \left[ 2 \cdot 16^p \sigma^{2p} L p C_f e^{2d} \left( 1 + \mathbb{1}_{p=2} \frac{\sqrt{L}}{22\sigma^2} \cdot \mathbb{E}[\|Z_1\|^8]^{1/4} \right) \cdot \right.$$

$$\left. \exp\left( - \left( \frac{L^{1/p}}{2^{1/p}} - \exp(\frac{2}{p}\sigma^2 \|Uz\|^2) \right) \cdot \frac{\mathbb{E}[\|f(Z_1)\|^4]^{1/2p}}{16\sigma^2 C_f^{2/p}} \right) \|g(z)\|^2 \mathbb{1}_{\|z\| \leq B} \right]$$

$$+ 2\mathbb{E}_z \left[ \mathbb{1}_{\|z\| > B} \left( \left\| \frac{R_L}{S_L} g(z) \right\|^2 + \left\| \frac{R}{S} g(z) \right\|^2 \right) \right]$$

$$\leq \mathbb{E}_z \left[ \frac{2C(z)\ln(L)}{L} \|g(z)\|^2 \mathbb{1}_{\|z\| \leq B} \right] + \mathbb{E}_z \left[ 2 \cdot 16^p \sigma^{2p} L p C_f e^{2d} \left( 1 + \mathbb{1}_{p=2} \frac{\sqrt{L}}{22\sigma^2} \cdot \mathbb{E}[\|Z_1\|^8]^{1/4} \right) \cdot \right.$$

$$\left. \exp\left( - \left( \frac{L^{1/p}}{2^{1/p}} - \exp(\frac{2}{p}\sigma^2 \|Uz\|^2) \right) \cdot \frac{\mathbb{E}[\|f(Z_1)\|^4]^{1/2p}}{16\sigma^2 C_f^{2/p}} \right) \|g(z)\|^2 \mathbb{1}_{\|z\| \leq B} \right]$$

$$+ 2\mathbb{E}_z \left[ \mathbb{1}_{\|z\| > B} \left( \mathbb{E}_{z_1,\ldots,z_L} \left[ \max_{\ell \in [L]} \|f(z_\ell)\|^2 \right] \|g(z)\|^2 + \frac{\|R \cdot g(z)\|^2}{S^2} \right) \right],$$

where for the last term, we used the deterministic bound $\|\frac{R_L}{S_L}\| \leq \max_{\ell \in [L]} \|f(z_\ell)\|$. We bound the three terms separately.

**Bound on Term 1.** By choosing $B = \frac{1}{\sigma \|U\|_{op}} \sqrt{\ln(L)/6}$,

$$\mathbb{1}_{\|z\| \leq B} C(z) = \mathbb{1}_{\|z\| \leq B} 108 \exp(4\sigma^2 \|Uz\|^2) \sqrt{\mathbb{E}_{Z_1}[\|f(Z_1)\|^4]}$$

$$\leq 108 \exp\left( \frac{2\ln(L)}{3} \right) \sqrt{\mathbb{E}_{Z_1}[\|f(Z_1)\|^4]}$$

where we used on the second line that, on the event $\{\|z\| \leq B\}$,

$$\sigma^2 \|Uz\|^2 \leq \sigma^2 \|U\|_{op}^2 B^2$$

$$= \frac{\ln(L)}{6} \, .$$

So for the first term, we have the bound

$$\mathbb{E}_z \left[ \frac{2C(z)\ln(L)}{L} \|g(z)\|^2 \mathbb{1}_{\|z\| \leq B} \right] \leq 216 \sqrt{\mathbb{E}_{Z_1}[\|f(Z_1)\|^4]} \mathbb{E}_z[\|g(z)\|^2] \frac{\ln(L)}{L^{1/3}}. \tag{13}$$

**Bound on Term 2.** Again, using that $\sigma^2 \|Uz\|^2 \leq \frac{\ln(L)}{6}$ on the event $\{\|z\| \leq B\}$, the exponential in the second term can be bounded as:

$$\exp\left( - \left( \frac{L^{1/p}}{2^{1/p}} - \exp(\frac{2}{p}\sigma^2 \|Uz\|^2) \right) \cdot \frac{\mathbb{E}[\|f(Z_1)\|^4]^{1/2p}}{16\sigma^2 C_f^{2/p}} \right) \mathbb{1}_{\|z\| \leq B}$$

$$\leq \exp\left( - \left( \frac{L^{1/p}}{2^{1/p}} - L^{1/3p} \right) \cdot \frac{\mathbb{E}[\|f(Z_1)\|^4]^{1/2p}}{16\sigma^2 C_f^{2/p}} \right),$$

so that the second term can be bounded by

$$\tilde{c}_1 \sigma^{2p} \mathbb{E}_z[\|g(z)\|^2] L^{3/2} \exp(-\frac{\tilde{c}_2}{\sigma^2} L^{1/p})$$

where $\tilde{c}_1, \tilde{c}_2$ are positive constants hiding only dependencies in $f$ and $d$. This matches the first term in the more explicit bound described by Equation (12). From now on, we show that this term is dominated by a polylog term as given in Lemma C.6. Note that for any $L, \sigma > 0$ and $p \in \{1, 2\}$

$$\tilde{c}_1 \sigma^{2p} \mathbb{E}_z[\|g(z)\|^2] L^{3/2} \exp(-\frac{\tilde{c}_2}{\sigma^2} L^{1/p}) \leq 5\tilde{c}_1 \sigma^{2p} \mathbb{E}_z[\|g(z)\|^2] L^{3/2} \left(\frac{\tilde{c}_2}{\sigma^2} L^{1/p}\right)^{-2p},$$

where we used that $e^{-x} \leq 5x^{-2p}$, when $x > 0$ and $p \in \{1, 2\}$. After simplifying, this finally yields that the second term can be bounded by $5\tilde{c}_1 \tilde{c}_2^{-2p} \sigma^{6p} L^{-1/2}$.

**Bound on Term 3.** First note that

$$\mathbb{E}_{z_1,\ldots,z_L}\left[\max_{\ell \in [L]} \|f(z_\ell)\|^4\right] \leq C_f^4 \cdot \mathbb{E}_{z_1,\ldots,z_L}\left[\max_{\ell \in [L]} \|z_\ell\|^{4p}\right].$$

By Cauchy-Schwarz inequality, we have

$$\mathbb{E}_z\left[\mathbb{1}_{\|z\|>B}\left(\mathbb{E}_{z_1,\ldots,z_L}[\max_{\ell \in [L]} \|f(z_\ell)g(z)\|^2] + \frac{\|R \cdot g(z)\|^2}{S^2}\right)\right]$$

$$\leq \sqrt{2\mathbb{E}_z\left[\|g(z)\|^4 \mathbb{E}_{z_1,\ldots,z_L}\left[\max_{\ell \in [L]} \|f(z_\ell)\|^4\right]\right] + 2\mathbb{E}_z\left[\frac{\|R \cdot g(z)\|^4}{S^4}\right]} \cdot \sqrt{\mathbb{P}(\|z\| \geq B)}$$

$$\leq \left(\sqrt{2\mathbb{E}_z[\|g(z)\|^4]\mathbb{E}_{z_1,\ldots,z_L}\left[\max_{\ell \in [L]} \|f(z_\ell)\|^4\right]} + \sqrt{2\mathbb{E}_z\left[\frac{\|R \cdot g(z)\|^4}{S^4}\right]}\right) \cdot \exp\left(d - \frac{B^2}{32}\right)$$

$$\leq \left(\sqrt{2\mathbb{E}_z[\|g(z)\|^4]\mathbb{E}_{z_1,\ldots,z_L}\left[\max_{\ell \in [L]} \|f(z_\ell)\|^4\right]} + \sqrt{2\mathbb{E}_z\left[\frac{\|R \cdot g(z)\|^4}{S^4}\right]}\right) \cdot \exp\left(d - \frac{\ln(L)}{192\sigma^2\|U\|_{\mathrm{op}}^2}\right)$$

$$\leq \left(\sigma^{2p}\sqrt{2\mathbb{E}_z[\|g(z)\|^4]C_f^4 C_{4p}(d^{2p} + \ln(L)^{2p})} + \sqrt{2\mathbb{E}_z\left[\frac{\|R \cdot g(z)\|^4}{S^4}\right]}\right) \cdot \exp\left(d - \frac{\ln(L)}{192\sigma^2\|U\|_{\mathrm{op}}^2}\right),$$

where we used Lemma D.2 for the last inequality. Now Assumption C.5 (ii) allows to bound $\sqrt{\mathbb{E}_z\left[\frac{\|R \cdot g(z)\|^4}{S^4}\right]}$ by $\sigma^{4p}\sqrt{c(f,g,U)}$ and to conclude. $\qquad\square$

## C.4. Concentration of the fourth order over input tokens (fixed query)

To control the concentration of the gradient, we will also need to derive an expected concentration bound similar to Lemma C.6, but with a power 4 instead of 2 in the concentration. In that goal, the two following sections derive results and analyses similar to Lemmas C.4 and C.6, but with fourth order moment. First, Lemma C.7 gives the fourth moment equivalent[2] of Lemma C.4.

**Lemma C.7.** *Assume $\mu$ is centered, $\sigma$ sub-Gaussian and there is $C_f \in \mathbb{R}_+$ such that $\sup_{z \in \mathbb{R}^d} \frac{\|f(z)\|}{\|z\|} \leq C_f$. Then for any $z \in \mathbb{R}^d$,*

$$\mathbb{E}_{z_1,\ldots,z_L}\left[\left\|\frac{\sum_{k=1}^{L}\exp(z_k^\top U z)f(z_k)}{\sum_{k=1}^{L}\exp(z_k^\top U z)} - \frac{\mathbb{E}_{Z_1}[\exp(Z_1^\top U z)f(Z_1)]}{\mathbb{E}_{Z_1}[\exp(Z_1^\top U z)]}\right\|^4\right] \leq \frac{2C\sqrt{\mathbb{E}[\|f(Z_1)\|^4]}}{\sqrt{L}} +$$

$$16^3 \sigma^4 L C_f^4 e^{2d}\left(1 + \frac{\sqrt{L}}{32\sqrt{2}\sigma^2} \cdot \mathbb{E}[\|Z_1\|^4]^{1/2}\right)$$

$$\cdot \exp\left(-\left(\frac{\sqrt{L}}{2\sqrt{2}} - \exp(2\sigma^2\|Uz\|^2)\right) \cdot \frac{\mathbb{E}[\|f(Z_1)\|^4]^{1/2}}{16\sigma^2 C_f^2}\right),$$

---

[2]Note that here the homogeneity degree $p$ is taken equal to 1, as it will be sufficient for our application.

$$C = C(U, f, z) = 108 \exp(4\sigma^2 \|Uz\|^2) \sqrt{\mathbb{E}_{Z_1}[\|f(Z_1)\|^4]}$$

*Proof.* Using the same notations as in the proof of Lemma C.1, it comes for $C$ defined by Corollary C.3, $t_0 = (\frac{16}{27})^2 \frac{C^2}{\delta_0^2 L^3}$:

$$\mathbb{E}_{z_1, \dots z_L}\left[\left\|\frac{R_L}{S_L} - \frac{R}{S}\right\|^4\right] = \int_0^\infty \mathbb{P}\left[\left\|\frac{R_L}{S_L} - \frac{R}{S}\right\|^4 \geq t\right] \mathrm{d}t$$

$$\leq \int_0^{t_0} \frac{C}{L\sqrt{t}} \mathrm{d}t + \int_{t_0}^\infty \mathbb{P}\left(\max_\ell \|f(z_\ell)\|^4 \geq \frac{t}{8} - \left\|\frac{R}{S}\right\|^4\right) \mathrm{d}t,$$

where we used Corollary C.3 for $t \leq t_0$, and for $t \geq t_0$ that

$$\left\|\frac{R_L}{S_L} - \frac{R}{S}\right\|^4 \geq t \qquad \Longrightarrow \qquad 8\left\|\frac{R_L}{S_L}\right\|^4 + 8\left\|\frac{R}{S}\right\|^4 \geq t$$

$$\Longrightarrow \qquad \left\|\frac{R_L}{S_L}\right\|^4 \geq \frac{t}{8} - \left\|\frac{R}{S}\right\|^4$$

$$\Longrightarrow \qquad \max_\ell \|f(z_\ell)\|^4 \geq \frac{t}{8} - \left\|\frac{R}{S}\right\|^4.$$

Therefore, processing the integration and applying a union bound again gives

$$\mathbb{E}_{z_1, \dots z_L}\left[\left\|\frac{R_L}{S_L} - \frac{R}{S}\right\|^4\right] \leq \frac{2C\sqrt{t_0}}{L} + 8L \int_{\left(\frac{t_0}{8} - \frac{\|R\|^4}{S^4}\right)_+}^\infty \mathbb{P}\left(C_f^4 \|z_1\|^4 \geq t\right) \mathrm{d}t$$

$$\leq \frac{2C\sqrt{t_0}}{L} + 8L \int_{\left(\frac{t_0}{8} - \frac{\|R\|^4}{S^4}\right)_+}^\infty \mathbb{P}\left(\|z_1\|^2 \geq \frac{t^{1/2}}{C_f^2}\right) \mathrm{d}t$$

$$\leq \frac{2C\sqrt{t_0}}{L} + 8Le^{2d} \int_{\left(\frac{t_0}{8} - \frac{\|R\|^4}{S^4}\right)_+}^\infty \exp\left(-\frac{t^{1/2}}{16\sigma^2 C_f^2}\right) \mathrm{d}t$$

$$\leq \frac{2C\sqrt{t_0}}{L} + 16Le^{2d} \int_{\left(\frac{t_0}{8} - \frac{\|R\|^4}{S^4}\right)_+^{1/2}}^\infty u \exp\left(-\frac{u}{16\sigma^2 C_f^2}\right) \mathrm{d}u,$$

where we use that for $\sigma$ sub-Gaussian random variables, for any $t$, $\mathbb{P}\left(\|z_1\|^2 \geq t\right) \leq e^{2d - t/16\sigma^2}$ (see Lemma D.2 in Lemma D). Moreover, note that

$$\int_{\left(\frac{t_0}{8} - \frac{\|R\|^4}{S^4}\right)_+^{1/2}}^\infty u \exp\left(-\frac{u}{16\sigma^2 C_f^2}\right) \mathrm{d}u = 16^2 \sigma^4 C_f^4 \int_{\left(\frac{t_0}{8} - \frac{\|R\|^4}{S^4}\right)_+^{1/2}/(16\sigma^2 C_f^2)}^\infty v \exp\left(-v\right) \mathrm{d}v$$

$$= 16^2 \sigma^4 C_f^4 \left(1 + \left(\frac{t_0}{8} - \frac{\|R\|^4}{S^4}\right)_+^{1/2}/(16\sigma^2 C_f^2)\right)$$

$$\cdot \exp\left(-\left(\frac{t_0}{8} - \frac{\|R\|^4}{S^4}\right)_+^{1/2}/(16\sigma^2 C_f^2)\right).$$

Recall that

$$t_0 = L\mathbb{E}[\|f(Z_1)\|^4] \qquad \text{and} \qquad \frac{\|R\|^4}{S^4} \leq \exp(4\sigma^2 \|Uz\|^2) \mathbb{E}[\|f(Z_1)\|^4].$$

By definition of $C_f$, $\mathbb{E}[\|f(Z_1)\|^4] \leq C_f^4 \mathbb{E}[\|Z_1\|^4]$, using these three last inequalities, we get that

$$\left(\frac{t_0}{8} - \frac{\|R\|^4}{S^4}\right)_+^{1/2} \cdot \frac{1}{16\sigma^2 C_f^2} \geq \left(\frac{L}{8} - \exp(4\sigma^2 \|Uz\|^2)\right)_+^{1/2} \cdot \frac{\mathbb{E}[\|f(Z_1)\|^4]^{1/2}}{16\sigma^2 C_f^2}$$

$$\text{and} \qquad \left(\frac{t_0}{8} - \frac{\|R\|^4}{S^4}\right)_+^{1/2} \cdot \frac{1}{16\sigma^2 C_f^2} \leq \frac{\sqrt{t_0}}{2\sqrt{2}} \cdot \frac{1}{16\sigma^2 C_f^2} \leq \frac{L^{1/2}}{32\sqrt{2}\sigma^2} \cdot \mathbb{E}[\|Z_1\|^4]^{1/2}.$$

Finally this yields

$$\mathbb{E}_{z_1,\dots z_L}\left[\left\|\frac{R_L}{S_L}-\frac{R}{S}\right\|^4\right] \le \frac{2C\sqrt{\mathbb{E}[\|f(Z_1)\|^4]}}{\sqrt{L}}+16^3\sigma^4 LC_f^4 e^{2d}\left(1+\frac{\sqrt{L}}{32\sqrt{2}\sigma^2}\cdot\mathbb{E}[\|Z_1\|^4]^{1/2}\right)$$
$$\cdot\exp\left(-\left(\frac{L}{8}-\exp(4\sigma^2\|Uz\|^2)\right)_+^{1/2}\cdot\frac{\mathbb{E}[\|f(Z_1)\|^4]^{1/2}}{16\sigma^2 C_f^2}\right),$$

which concludes the proof by subadditivity of the square root. $\qquad\square$

## C.5. Concentration of the fourth order over both query and input tokens

Assumption C.8 and Lemma C.9 below then correspond to the fourth moment equivalent of Assumption C.5 and Lemma C.6, with $p=1$.

**Assumption C.8.** *Assume that $f,g$ satisfy the following properties for the family of probability distributions $P\subseteq\mathcal{P}(\mathbb{R}^d)$:*

*(i) there is $C_f\in\mathbb{R}_+$ such that $\sup_{z\in\mathbb{R}^d}\frac{\|f(z_1)\|}{\|z_1\|}\le C_f$;*

*(ii) for any $U\in\mathbb{R}^{d\times d}$ and $\mu\in P$ that is $\sigma$ sub-Gaussian, $\mathbb{E}_z\left[\frac{\|\mathbb{E}_{Z_1}[\exp(Z_1^\top Uz)f(Z_1)g(z)]\|^8}{\mathbb{E}_{Z_1}[\exp(Z_1^\top Uz)]^8}\right]\le\sigma^{16}c(f,g,U)<+\infty$;*

*(iii) $\mathbb{E}_z[\|g(z)\|^8]<+\infty$.*

Again, we could use a more general scaling in $\sigma$ for Assumption C.8 (ii).

**Lemma C.9** (General finite-length bound). *Assume that $\mu$ and $\nu$ are both centered, respectively $\sigma$ and $1$ sub-Gaussian distributions with $\sigma\ge 1$. Moreover, consider functions $f,g$ satisfying Assumption C.8 with $\mu\in P$, assuming that the product $f(z_1)g(z)$ is legit and that $\|f(z_1)g(z)\|\le\|f(z_1)\|\,\|g(z)\|$. Then there exist constants $c_1,c_2>0$ depending solely on $d,f,g,P,U$ such that,*

$$\mathbb{E}\left[\left\|\frac{\sum_{k=1}^L\exp(z_{\text{query}}^\top U^\top z_k)f(z_k)g(z)}{\sum_{k=1}^L\exp(z_{\text{query}}^\top U^\top z_k)}-\frac{\mathbb{E}_{Z_1}[\exp(z_{\text{query}}^\top U^\top Z_1)f(Z_1)g(z)]}{\mathbb{E}_{Z_1}[\exp(z_{\text{query}}^\top U^\top Z_1)]}\right\|^4_{L^4(\nu)}\right]\le c_1\sigma^{12}\frac{\ln(L)^2}{L^{c_2/\sigma^2}}$$

*where $Z_1\sim\mu$ and the expectation is taken over $(z_1,\dots,z_L)\sim\mu^{\otimes L}$.*

The proof provides a more explicit bound of the form

$$\mathbb{E}\left[\left\|\frac{\sum_{k=1}^L\exp(z_{\text{query}}^\top U^\top z_k)f(z_k)}{\sum_{k=1}^L\exp(z_{\text{query}}^\top U^\top z_k)}-\frac{\mathbb{E}_{Z_1}[\exp(z_{\text{query}}^\top U^\top Z_1)f(Z_1)]}{\mathbb{E}_{Z_1}[\exp(z_{\text{query}}^\top U^\top Z_1)]}\right\|^4_{L^4(\nu)}\right] \tag{14}$$
$$\le\tilde{c}_1\sigma^4 L^{3/2}\exp(-\tilde{c}_2\sqrt{L}/\sigma^2)+\tilde{c}_3\frac{1}{L^{1/6}}+\tilde{c}_4\sigma^8\frac{\ln(L)^2}{L^{1/(384\sigma^2\|U\|_{\text{op}}^2)}},$$

where $\tilde{c}_1,\tilde{c}_2,\tilde{c}_3,\tilde{c}_4$ are positive constants depend solely on $d,f,g,P,U$.

*Proof.* Again in this section, we follow the notation used in the proofs of Lemmas C.1 and C.7. To get a bound on the

expectation, we split the computation as follows for some $B \in \mathbb{R}_+$

$$\mathbb{E}_z \left[ \mathbb{E}_{z_1,\ldots,z_L} \left[ \left\| \frac{R_L}{S_L} g(z) - \frac{R}{S} g(z) \right\|^4 \right] \right]$$

$$\leq \mathbb{E}_z \left[ \mathbb{E}_{z_1,\ldots,z_L} \left[ \left\| \frac{R_L}{S_L} - \frac{R}{S} \right\|^4 \right] \|g(z)\|^4 \mathbb{1}_{\|z\| \leq B} \right] + \mathbb{E}_z \left[ \mathbb{E}_{z_1,\ldots,z_L} \left[ \left\| \frac{R_L}{S_L} g(z) - \frac{R}{S} g(z) \right\|^4 \right] \mathbb{1}_{\|z\| > B} \right]$$

$$\leq \mathbb{E}_z \left[ \frac{2C(z)\sqrt{\mathbb{E}[\|f(Z_1)\|^4]}}{\sqrt{L}} \|g(z)\|^4 \mathbb{1}_{\|z\| \leq B} \right] + \mathbb{E}_z \left[ 16^3 \sigma^4 L C_f^4 e^{2d} \left( 1 + \frac{\sqrt{L}}{32\sqrt{2}\sigma^2} \cdot \mathbb{E}[\|Z_1\|^4]^{1/2} \right) \cdot \right.$$

$$\left. \exp\left( -\left( \frac{\sqrt{L}}{2\sqrt{2}} - \exp(2\sigma^2\|Uz\|^2) \right) \cdot \frac{\mathbb{E}[\|f(Z_1)\|^4]^{1/2}}{16\sigma^2 C_f^2} \right) \|g(z)\|^4 \mathbb{1}_{\|z\| \leq B} \right]$$

$$+ 8\mathbb{E}_z \left[ \mathbb{1}_{\|z\| > B} \left( \left\| \frac{R_L}{S_L} g(z) \right\|^4 + \left\| \frac{R}{S} g(z) \right\|^4 \right) \right]$$

$$\leq \mathbb{E}_z \left[ \frac{2C(z)\sqrt{\mathbb{E}[\|f(Z_1)\|^4]}}{\sqrt{L}} \|g(z)\|^4 \mathbb{1}_{\|z\| \leq B} \right] + \mathbb{E}_z \left[ 16^3 \sigma^4 L C_f^4 e^{2d} \left( 1 + \frac{\sqrt{L}}{32\sqrt{2}\sigma^2} \cdot \mathbb{E}[\|Z_1\|^4]^{1/2} \right) \cdot \right.$$

$$\left. \exp\left( -\left( \frac{\sqrt{L}}{2\sqrt{2}} - \exp(2\sigma^2\|Uz\|^2) \right) \cdot \frac{\mathbb{E}[\|f(Z_1)\|^4]^{1/2}}{16\sigma^2 C_f^2} \right) \|g(z)\|^4 \mathbb{1}_{\|z\| \leq B} \right]$$

$$+ 8\mathbb{E}_z \left[ \mathbb{1}_{\|z\| > B} \left( \mathbb{E}_{z_1,\ldots,z_L} [\max_{\ell \in [L]} \|f(z_\ell)\|^4] \|g(z)\|^4 + \frac{\|R \cdot g(z)\|^4}{S^4} \right) \right],$$

where for the last term, we used the deterministic bound $\|\frac{R_L}{S_L}\| \leq \max_{\ell \in [L]} \|f(z_\ell)\|$. We now bound the three terms separately.

**Bound on Term 1.** By choosing $B = \frac{1}{\sigma\|U\|_{\mathrm{op}}} \sqrt{\ln(L)/12}$,

$$\mathbb{1}_{\|z\| \leq B} C(U, f, z) = \mathbb{1}_{\|z\| \leq B} 108 \exp(4\sigma^2\|Uz\|^2)\sqrt{\mathbb{E}_{Z_1}[\|f(Z_1)\|^4]}$$

$$\leq 108 \exp\left( \frac{\ln(L)}{3} \right) \sqrt{\mathbb{E}_{Z_1}[\|f(Z_1)\|^4]}$$

where we used on the second line that, on the event $\{\|z\| \leq B\}$,

$$\sigma^2\|Uz\|^2 \leq \sigma^2\|U\|_{\mathrm{op}}^2 B^2$$

$$= \frac{\ln(L)}{12} .$$

So for the first term, we have the bound

$$\mathbb{E}_z \left[ \frac{2C(z)\sqrt{\mathbb{E}[\|f(Z_1)\|^4]}}{\sqrt{L}} \|g(z)\|^4 \mathbb{1}_{\|z\| \leq B} \right] \leq 216\mathbb{E}_{Z_1}[\|f(Z_1)\|^4]\mathbb{E}_z[\|g(z)\|^4]\frac{1}{L^{1/6}}. \tag{15}$$

**Bound on Term 2.** Again, using that $\sigma^2\|Uz\|^2 \leq \frac{\ln(L)}{12}$ on the event $\{\|z\| \leq B\}$, the exponential in the second term can be bounded as:

$$\exp\left( -\left( \frac{\sqrt{L}}{2\sqrt{2}} - \exp(2\sigma^2\|Uz\|^2) \right) \cdot \frac{\mathbb{E}[\|f(Z_1)\|^4]^{1/2}}{16\sigma^2 C_f^2} \right) \mathbb{1}_{\|z\| \leq B}$$

$$\leq \exp\left( -\left( \frac{\sqrt{L}}{2\sqrt{2}} - L^{1/6} \right) \cdot \frac{\mathbb{E}[\|f(Z_1)\|^4]^{1/2}}{16\sigma^2 C_f^2} \right),$$

so that the second term can be bounded by

$$\tilde{c}_1 \sigma^4 \mathbb{E}_z[\|g(z)\|^4] L^{3/2} \exp(-\frac{\tilde{c}_2}{\sigma^2}\sqrt{L})$$

where $\tilde{c}_1, \tilde{c}_2$ are positive constants hiding only dependencies in $f$ and $d$. This matches the first term in the more explicit bound described by Equation (12). From now on, we can show this term is dominated by a polylog term in $L$ as in the proof Lemma C.6, so that the second term can be bounded by $5\tilde{c}_1\tilde{c}_2^{-4}\sigma^{12}L^{-1/2}$.

**Bound on Term 3.** First note that

$$\mathbb{E}_{z_1,\dots,z_L}\left[\max_{\ell\in[L]}\|f(z_\ell)\|^8\right] \le C_f^8 \cdot \mathbb{E}_{z_1,\dots,z_L}\left[\max_{\ell\in[L]}\|z_\ell\|^8\right]$$

By Cauchy-Schwarz inequality, we have

$$\mathbb{E}_z\left[\mathbb{1}_{\|z\|>B}\left(\mathbb{E}_{z_1,\dots,z_L}[\max_{\ell\in[L]}\|f(z_\ell)g(z)\|^4] + \frac{\|R\cdot g(z)\|^4}{S^4}\right)\right]$$

$$\le \sqrt{2\mathbb{E}_z\left[\|g(z)\|^8\mathbb{E}_{z_1,\dots,z_L}\left[\max_{\ell\in[L]}\|f(z_\ell)\|^8\right]\right] + 2\mathbb{E}_z\left[\frac{\|R\cdot g(z)\|^8}{S^8}\right]} \cdot \sqrt{\mathbb{P}(\|z\| \ge B)}$$

$$\le \left(\sqrt{2\mathbb{E}_z[\|g(z)\|^8]\mathbb{E}_{z_1,\dots,z_L}\left[\max_{\ell\in[L]}\|f(z_\ell)\|^8\right]} + \sqrt{2\mathbb{E}_z\left[\frac{\|R\cdot g(z)\|^8}{S^8}\right]}\right)\cdot\exp\left(d-\frac{B^2}{32}\right)$$

$$\le \left(\sqrt{2\mathbb{E}_z[\|g(z)\|^8]\mathbb{E}_{z_1,\dots,z_L}\left[\max_{\ell\in[L]}\|f(z_\ell)\|^8\right]} + \sqrt{2\mathbb{E}_z\left[\frac{\|R\cdot g(z)\|^8}{S^8}\right]}\right)\cdot\exp\left(d-\frac{\ln(L)}{384\sigma^2\|U\|_{\mathrm{op}}^2}\right)$$

$$\le \left(\sigma^4\sqrt{2\mathbb{E}_z[\|g(z)\|^8]C_f^8 C_8(d^4+\ln(L)^4)} + \sqrt{2\mathbb{E}_z\left[\frac{\|R\cdot g(z)\|^8}{S^8}\right]}\right)\cdot\exp\left(d-\frac{\ln(L)}{384\sigma^2\|U\|_{\mathrm{op}}^2}\right),$$

where we used Lemma D.2 for the last inequality. Assumption C.8 (ii) then ensures the bound $\sqrt{\mathbb{E}_z\left[\frac{\|R\cdot g(z)\|^8}{S^8}\right]} \le \sigma^8\sqrt{c(f,g,U)}$. $\qquad\square$

### C.6. Proof of Proposition 3.1

The proof of Proposition 3.1 consists in the application of Lemma C.6 when setting $f(z_1) = Vz_1$ for a given value matrix $V$ and $g(z) = 1$. Therefore, we have to verify that assumptions of Lemma C.6 are met for such a choice of $f$.

Note that, for $p = 1$, $f$ satisfies Assumption C.5(i) with $C_f = \|V\|_{\mathrm{op}}$. Assumption C.5(ii) is a consequence of Lemma D.1. Indeed for any $U \in \mathbb{R}^{d\times d}$ and the above $f$,

$$\mathbb{E}_Z\frac{\left\|\mathbb{E}_{Z_1}\left[\exp(Z_1^\top UZ)f(Z_1)\right]\right\|^4}{\mathbb{E}_{Z_1}\left[\exp(Z_1^\top UZ)\right]^4} = \|T^{U,V}[\mu]\|_{L^4(\nu)}^4 \le 4^4\sigma^8\|V\|_{\mathrm{op}}^4\|U\|_{\mathrm{op}}^4 d^2,$$

by Lemma D.1. Finally, note that Assumption C.5(iii) is automatic for the given choice of $g$.

### C.7. Proof of Proposition 3.4

To derive the analysis, we derive the computations "row-wise" in $V$, i.e., we study $T^{U,v_i}[\mu](z) \in \mathbb{R}$ where $v_i$ is the $i$-th row of matrix $V$. The row-wise gradients are given as follows

$$\left(\nabla_V T^{U,V}[\hat{\mu}](z)\right)_i = \frac{\sum_{k=1}^L \exp(z_k^\top Uz)z_k}{\sum_{k=1}^L \exp(z_k^\top Uz)}$$

$$\left(\nabla_U T^{U,V}[\hat{\mu}](z)\right)_i = \frac{\sum_{k=1}^L \exp(z_k^\top Uz)(v_i^\top z_k)(z_k^\top z)}{\sum_{k=1}^L \exp(z_k^\top Uz)} - \frac{\sum_{k=1}^L \exp(z_k^\top Uz)(v_i^\top z_k)}{\sum_{k=1}^L \exp(z_k^\top Uz)}\frac{\sum_{k=1}^L \exp(z_k^\top Uz)(z_k^\top z)}{\sum_{k=1}^L \exp(z_k^\top Uz)}$$

i) The first inequality of Proposition 3.4 (on $\left(\nabla_V T^{U,V}[\hat{\mu}](z)\right)_i$) comes from Lemma C.6 with $f = \mathrm{id}$ and $g = 1$. Note that, for $p = 1$, $f$ satisfies Assumption C.5(i) with $C_f = 1$. Assumption C.5(ii) is a rewriting of the moment assumption, indeed for any $U \in \mathbb{R}^{d \times d}$ and the above $f$,

$$
\mathbb{E}_Z \frac{\left\| \mathbb{E}_{Z_1 \sim \mu} \left[ \exp(Z_1^\top U Z) f(Z_1) \right] \right\|^4}{\mathbb{E}_{Z_1} \left[ \exp(Z_1^\top U Z) \right]^4} \leq \mathbb{E}_Z \frac{\mathbb{E}_{Z_1} \left[ \exp(Z_1^\top U Z) \| f(Z_1) \|^4 \right]}{\mathbb{E}_{Z_1} \left[ \exp(Z_1^\top U Z) \right]^4} \tag{16}
$$
$$
= \mathbb{E}_Z \frac{\mathbb{E}_{Z_1} \left[ \exp(Z_1^\top U Z) \| Z_1 \|^4 \right]}{\mathbb{E}_{Z_1} \left[ \exp(Z_1^\top U Z) \right]^4} \leq \sigma^8 M_{4,0},
$$

by assumption. Finally, note that Assumption C.5(iii) is automatic for the given choice of $g$.

ii) Regarding $\left(\nabla_U T^{U,V}[\hat{\mu}](z)\right)_i$, let us first give a concentration on the first term. In particular, we can show a bound

$$
\mathbb{E}_Z \left[ \left\| \frac{\sum_{k=1}^L \exp(z_k^\top U z)(v_i^\top z_k)(z_k^\top z)}{\sum_{k=1}^L \exp(z_k^\top U z)} - \frac{\mathbb{E}_{Z_1}[\exp(z_k^\top U z)(v_i^\top Z_1)(Z_1^\top z)]}{\mathbb{E}_{Z_1}[\exp(Z_1^\top U z)]} \right\|^2 \right] \leq c_1 \sigma^{12} \frac{\ln(L)^2}{L^{c_2/\sigma^2}}
$$

using again Lemma C.6, with $f(z_k) = (v_i^\top z_k) z_k$, $p = 2$ and $g(z) = z^\top$. Indeed, for $p = 2$, $f$ satisfies Assumption C.5(i) with $C_f = \|v_i\|$. Assumption C.5(ii) comes from the moment assumption. Indeed for any $U \in \mathbb{R}^{d \times d}$, the above $f$ and $C_f$, we have similarly to Equation (16),

$$
\mathbb{E}_Z \frac{\left\| \mathbb{E}_{Z_1} \left[ \exp(Z_1^\top U Z) f(Z_1) g(Z) \right] \right\|^4}{\mathbb{E}_{Z_1} \left[ \exp(Z_1^\top U Z) \right]^4} \leq \|v_i\|^4 \mathbb{E}_Z \frac{\mathbb{E}_{Z_1} \left[ \exp(Z_1^\top U Z) \| Z_1 \|^8 \| Z \|^4 \right]}{\mathbb{E}_{Z_1} \left[ \exp(Z_1^\top U Z) \right]} \leq \sigma^{16} M_{8,4}.
$$

Finally, note that Assumption C.5(iii) comes from the sub-Gaussian property.

It now remains to control the second term in $\left(\nabla_U T^{U,V}[\hat{\mu}](z)\right)_i$. It is the product of two terms that we can call

$$
\hat{X} = \frac{\sum_{k=1}^L \exp(z_k^\top U z)(v_i^\top z_k)}{\sum_{k=1}^L \exp(z_k^\top U z)} \qquad \text{and} \qquad \hat{Y} = \frac{\sum_{k=1}^L \exp(z_k^\top U z) z_k z^\top}{\sum_{k=1}^L \exp(z_k^\top U z)}.
$$

We also denote by $X, Y$ their infinite counterparts, i.e.,

$$
X = \frac{\mathbb{E}_{Z_1}[\exp(Z_1^\top U z)(v_i^\top Z_1)]}{\mathbb{E}_{Z_1}[\exp(Z_1^\top U z)]} \qquad \text{and} \qquad Y = \frac{\mathbb{E}_{Z_1}[\exp(Z_1^\top U z) Z_1 z^\top]}{\mathbb{E}_{Z_1}[\exp(z_k^\top U z)]}.
$$

Remark that

$$
\mathbb{E} \left[ \| \hat{X}\hat{Y} - XY \|_F^2 \right] \leq 2\mathbb{E} \left[ \| \hat{X}(\hat{Y} - Y) \|_F^2 + \| Y(\hat{X} - X) \|_F^2 \right]
$$
$$
\leq 2 \left( \mathbb{E}[\hat{X}^4] \mathbb{E}[\| \hat{Y} - Y \|^4] \right)^{1/2} + 2 \left( \mathbb{E}[\| Y \|^4] \mathbb{E}[(\hat{X} - X)^4] \right)^{1/2}
$$
$$
\leq 2 \left( 8 (\mathbb{E}[X^4] + \mathbb{E}[(\hat{X} - X)^4]) \mathbb{E}[\| \hat{Y} - Y \|^4] \right)^{1/2} + 2 \left( \mathbb{E}[\| Y \|^4] \mathbb{E}[(\hat{X} - X)^4] \right)^{1/2}. \tag{17}
$$

Similarly to Equation (16), first note that

$$
\mathbb{E}[X^4] \leq \|v_i\|^4 \sigma^8 M_{4,0} \qquad \text{and} \qquad \mathbb{E}[\| Y \|^4] \leq \sigma^8 M_{4,4}.
$$

Moreover, we can use Lemma C.9 to bound both $\mathbb{E}[(\hat{X} - X)^4]$ and $\mathbb{E}[\| \hat{Y} - Y \|^4]$. Indeed, the former can be bounded by applying Lemma C.9 with $f(z_1) = v_i^\top z_1$ and $g(z) = 1$. Assumption C.8 (i) is then satisfied for $C_f = \|v_i\|$, Assumption C.8 (iii) is automatic and Assumption C.8 (ii) comes from the moment assumption:

$$
\mathbb{E}_z \left[ \frac{\left\| \mathbb{E}_{Z_1 \sim \mu} \left[ \exp(Z_1^\top U z) f(Z_1) g(z) \right] \right\|^8}{\mathbb{E}_{Z_1} \left[ \exp(Z_1^\top U z) \right]^8} \right] = \mathbb{E}_z \left[ \| \mathbb{E}_{Z_1 \sim \mu_z} [f(Z_1) g(z)] \|^8 \right]
$$
$$
\leq \|v_i\|^8 \mathbb{E}_z \left[ \mathbb{E}_{Z_1 \sim \mu_z} \left[ \| Z_1 \|^8 \right] \right] \leq \|v_i\|^8 \sigma^{16} M_{8,0}.
$$

Similarly, $\mathbb{E}[\|\hat{Y} - Y\|^4]$ is bounded by applying Lemma C.9 with $f(z_1) = z_1$ and $g(z) = z^\top$. Again, Assumption C.8 holds with

$$\mathbb{E}_z\left[\frac{\left\|\mathbb{E}_{Z_1 \sim \mu}\left[\exp(Z_1^\top U z)f(Z_1)g(z)\right]\right\|^8}{\mathbb{E}_{Z_1}\left[\exp(Z_1^\top U z)\right]^8}\right] \leq \sigma^{16} M_{8,8}.$$

These concentrations along with Equation (17) yield for constants $c_1, c_2$ depending solely on $d, V, U$ that

$$\mathbb{E}\left[\|\hat{X}\hat{Y} - XY\|_F^2\right] \leq c_1 \sigma^{12} \frac{\ln(L)^2}{L^{c_2/\sigma^2}},$$

which finally allows to conclude.

## D. Auxiliary Lemmas

### D.1. Bounded moment for infinite-prompt output

**Lemma D.1.** *Assume $\mu$ and $\nu$ are respectively $\sigma$ and $1$ sub-Gaussian, centered measures. Then*

$$\|T^{K,Q,V}[\mu]\|_{L^4(\nu)} \leq 4\sigma^2 \|V\|_{\mathrm{op}} \|K^\top Q\|_{\mathrm{op}} \sqrt{d}.$$

*Proof.* The first parf of the proof (until Equation (18)) follows arguments similar to the one of Bobkov and Götze (2025, Proposition 2.4).

Define $\Psi(z) = \log \mathbb{E}_{z_1 \sim \mu}\left[e^{z^\top z_1}\right]$. By $\sigma$ sub-Gaussianity of $z_1$, $\Psi(z) \leq \frac{\sigma^2}{2}\|z\|^2$. So we can define the non-negative function $\varphi: z \mapsto \frac{\sigma^2}{2}\|z\|^2 - \Psi(z)$. By Jensen inequality, $\Psi(z) \geq \mathbb{E}_{z_1}\left[z^\top z_1\right] = 0$, so that $\varphi(z) \leq \frac{\sigma^2}{2}\|z\|^2$.

As $\Psi$ is convex, it also holds that $\nabla^2 \varphi \preceq \sigma^2 \mathrm{I}_d$. So that a Taylor expansion yields for $h, z$

$$\varphi(z + h) \leq \varphi(z) + \langle \nabla\varphi(z), h\rangle + \frac{\sigma^2}{2}\|h\|^2.$$

In particular, as $\varphi(z + h) \geq 0$,

$$\varphi(z) \geq -\langle\nabla\varphi(z), h\rangle - \frac{\sigma^2}{2}\|h\|^2.$$

Taking $h = -\sigma^{-2}\nabla\varphi(z)$, it becomes

$$\frac{\sigma^2}{2}\|z\|^2 \geq \varphi(z) \geq \frac{1}{2\sigma^2}\|\nabla\varphi(z)\|^2, \tag{18}$$

i.e., $\|\nabla\varphi(z)\| \leq \sigma^2\|z\|$. By triangle inequality, we then have $\|\nabla\Psi(z)\| \leq 2\sigma^2\|z\|$.

From there, we can conclude by noting that $T^{K,Q,V}[\mu](z) = V\nabla\Psi(Q^\top K z)$, so that

$$\begin{aligned}
\|T^{K,Q,V}[\mu]\|_{L^4(\nu)} &\leq \|V\|_{\mathrm{op}}\mathbb{E}_{z \sim \nu}\left[\|\nabla\Psi(Q^\top K z)\|^4\right]^{1/4} \\
&\leq \|V\|_{\mathrm{op}}\mathbb{E}_{z \sim \nu}\left[\left(2\sigma^2\|Q^\top K z\|\right)^4\right]^{1/4} \\
&\leq 2\sigma^2\|V\|_{\mathrm{op}}\|K^\top Q\|_{\mathrm{op}}\mathbb{E}_{z \sim \nu}\left[\|z\|^4\right]^{1/4} \\
&\leq 4\sigma^2\|V\|_{\mathrm{op}}\|K^\top Q\|_{\mathrm{op}}\sqrt{d}.
\end{aligned}$$

$\square$

### D.2. Maximal Inequality

**Lemma D.2.** *Suppose $z_1, \ldots, z_L$ are $L$ i.i.d. $1$ sub-Gaussian random variables in $\mathbb{R}^d$. Then*

1. *for any $t \geq 0$, $\mathbb{P}\left(\|z_1\| \geq t\right) \leq e^{2d - t^2/16}$;*

2. *for any $p \geq 2$, there exists a constant $C_p \in \mathbb{R}$ such that $\mathbb{E}[\max_{\ell=1,\dots} \|z_\ell\|^p] \leq C_p(d^{p/2} + \ln(L)^{p/2})$.*

*Proof.* 1) For any $\delta \in (0,1)$, Theorem 1.19 by Rigollet and Hütter (2023) yields that

$$\mathbb{P}\left(\|z_1\| \geq 4\sqrt{d} + 2\sqrt{2\ln(1/\delta)}\right) \leq \delta.$$

Thus for any $t > 4\sqrt{d}$, with the transformation $t = 4\sqrt{d} + 2\sqrt{2\ln(1/\delta)}$, it rewrites:

$$\mathbb{P}\left(\|z_1\| \geq t\right) \leq e^{-\frac{1}{8}(t - 4\sqrt{d})^2}.$$

Using that for any $a, b \geq 0$, $(a-b)^2 \geq \frac{1}{2}a^2 - b^2$, this yields for any $t > 4\sqrt{d}$

$$\mathbb{P}\left(\|z_1\| \geq t\right) \leq e^{-\frac{t^2}{16} + 2d}.$$

Now noting that $e^{-\frac{t^2}{16} + 2d} \geq 1$ for any $t \in [0, 4\sqrt{d}]$ concludes the first point of Lemma D.2.

2) Take in the following $B \geq 1$. We can decompose the expectation as follows:

$$\mathbb{E}\left[\max_\ell \|z_\ell\|^p\right] = B + \int_B^\infty \mathbb{P}\left(\max_\ell \|z_\ell\|^p \geq t\right) \mathrm{d}t$$

$$\leq B + L \int_B^\infty \mathbb{P}\left(\|z_1\| \geq t^{1/p}\right) \mathrm{d}t$$

$$\leq B + L e^{2d} \int_B^\infty e^{-t^{2/p}/16} \mathrm{d}t,$$

where we used the first point of Lemma D.2 for the last inequality. From there, using change of variables, it comes:

$$\int_B^\infty e^{-t^{2/p}/16} \mathrm{d}t = \frac{p}{2}(16)^{p/2} \int_{\frac{B^{2/p}}{16}}^\infty u^{p/2-1} e^{-u} \mathrm{d}u$$

$$= \frac{p}{2}(16)^{p/2} \cdot \Gamma\left(\frac{p}{2}, \frac{B^{2/p}}{16}\right),$$

where we recall $\Gamma(\cdot, \cdot)$ is the upper incomplete Gamma function. Typical upper bounds on the upper incomplete Gamma function (Pinelis, 2020) show the existence of a constant $c_p$ such that for any $x \geq 1/16$, $\Gamma\left(\frac{p}{2}, x\right) \leq c_p e^{-x} x^{p/2}$, so that

$$\mathbb{E}\left[\max_\ell \|z_\ell\|^p\right] \leq B + L e^{2d} \frac{p}{2}(16)^{p/2} \cdot c_p \exp\left(-\frac{B^{2/p}}{16}\right) \frac{B}{16^{p/2}}.$$

Taking $B = 64^{p/2} d^{p/2} + 32^{p/2} \ln(L)^{p/2}$, note that, as $2/p \leq 1$, $B^{2/p} \geq \frac{1}{2}(64d + 32\ln(L))$, so that for $B \geq 1$

$$\mathbb{E}\left[\max_\ell \|z_\ell\|^p\right] \leq B + L e^{2d} \frac{p}{2} \cdot c_p \exp\left(-2d - \ln(L)\right) B$$

$$\leq \left(1 + \frac{p \cdot c_p}{2}\right) B.$$

Lemma D.2 then follows by plugging the value of $B$. $\qquad\square$

## E. Proofs of Section 3

### E.1. Proof of Theorem 4.3

Let $\varepsilon > 0$, and choose $T(\varepsilon)$ such that

$$\mathcal{R}_\infty^{\mathrm{ICL}}(T(\varepsilon)) \leq \lim_{t \to +\infty} \mathcal{R}_\infty^{\mathrm{ICL}}(t) + \varepsilon.$$

Recall that $\begin{pmatrix} U_\infty(t) \\ V_\infty(t) \end{pmatrix} \in B_\rho$ for any $t \geq 0$ by assumption. Using Lemma E.1, we can then choose $L^\star(\varepsilon)$ such that for any $L \geq L^\star(\varepsilon)$ and $t \in [0, T(\varepsilon)]$,

$$\left\| \begin{pmatrix} U_L(t) \\ V_L(t) \end{pmatrix} - \begin{pmatrix} U_\infty(t) \\ V_\infty(t) \end{pmatrix} \right\| \leq \rho.$$

In particular, $\begin{pmatrix} U_L(t) \\ V_L(t)) \end{pmatrix} \in B_{2\rho}$. Using Proposition E.3 and Lemma E.1, for any $t \in [0, T(\varepsilon)]$ and $L \geq L^\star(\varepsilon)$,

$$\mathcal{R}_L^{\mathrm{ICL}}(U_L(t), V_L(t)) - \mathcal{R}_\infty^{\mathrm{ICL}}(U_\infty(t), V_\infty(t)) = \mathcal{R}_L^{\mathrm{ICL}}(U_L(t), V_L(t)) - \mathcal{R}_\infty^{\mathrm{ICL}}(U_L(t), V_L(t))$$
$$+ \mathcal{R}_\infty^{\mathrm{ICL}}(U_L(t), V_L(t)) - \mathcal{R}_\infty^{\mathrm{ICL}}(U_\infty(t), V_\infty(t))$$
$$\leq g_1(L) + \kappa_\infty^{\mathrm{ICL}} \left\| \begin{pmatrix} U_L(t) \\ V_L(t) \end{pmatrix} - \begin{pmatrix} U_\infty(t) \\ V_\infty(t) \end{pmatrix} \right\|$$
$$\leq g_1(L) + \kappa_\infty^{\mathrm{ICL}} g_2(L) \cdot t \cdot \exp\left(\beta_L^{\mathrm{ICL}} t\right)$$

where $\kappa_\infty^{\mathrm{ICL}}$ is the Lipschitz constant of $\mathcal{R}_\infty^{\mathrm{ICL}}$ on $B_{2\rho}$. As $g_1$ and $g_2$ both go to 0 as $L \to \infty$, we can update $L^\star(\varepsilon)$ large enough so that for any $L \geq L^\star(\varepsilon)$ and $t \in [0, T(\varepsilon)]$,

$$\mathcal{R}_L^{\mathrm{ICL}}(U_L(t), V_L(t)) \leq \mathcal{R}_\infty^{\mathrm{ICL}}(U_\infty(t), V_\infty(t)) + \varepsilon.$$

In particular for $t = T(\varepsilon)$,

$$\mathcal{R}_L^{\mathrm{ICL}}(U_L(T(\varepsilon)), V_L(T(\varepsilon))) \leq \lim_{t\to\infty} \mathcal{R}_\infty^{\mathrm{ICL}}(U_\infty(t), V_\infty(t)) + 2\varepsilon.$$

As the risk is non-increasing along the flow, we finally get

$$\lim_{t\to\infty} \mathcal{R}_L^{\mathrm{ICL}}(U_L(t), V_L(t)) \leq \lim_{t\to\infty} \mathcal{R}_\infty^{\mathrm{ICL}}(U_\infty(t), V_\infty(t)) + 2\varepsilon.$$

### E.2. Technical lemmas

**Lemma E.1.** *Assume that $\ell$ is 1-smooth, $\nabla_1\ell(0,0) = 0$, $\mathcal{R}_\infty^{\mathrm{ICL}}$ is $C^2$, and that the gradient flow $\begin{pmatrix} U_\infty(t) \\ V_\infty(t) \end{pmatrix}$ is contained within the centered ball $B_\rho$ of radius $\rho$. Consider in addition that Assumption 4.2 holds.*

*Then for any $T$, there exists $L^\star$ such that for all $L \geq L^\star$ and for all $t \in [0, T]$,*

$$\left\| \begin{pmatrix} U_L(t) \\ V_L(t) \end{pmatrix} - \begin{pmatrix} U_\infty(t) \\ V_\infty(t) \end{pmatrix} \right\| \leq g_2(L) \cdot t \cdot \exp\left(\beta_L^{\mathrm{ICL}} t\right) \tag{19}$$

*where $g_2(L) \xrightarrow[L\to\infty]{} 0$.*

*Proof.* In the proof, for convenience, we set $\theta_L = \begin{pmatrix} U_L \\ V_L \end{pmatrix}$ and $\theta_\infty = \begin{pmatrix} U_\infty(t) \\ V_\infty(t) \end{pmatrix}$. We also define $\tau_L = \inf\{t \geq 0 : \theta_L(t) \notin B_{2\rho}\}$. Remark that for any $t \leq \tau_L$, $\theta_L(t) \in B_{2\rho}$, so that

$$\|\dot{\theta}_L(t) - \dot{\theta}_\infty(t)\| \leq \| - \nabla\mathcal{R}_L^{\mathrm{ICL}}(\theta_L(t)) + \nabla\mathcal{R}_\infty^{\mathrm{ICL}}(\theta_L(t))\| + \| - \nabla\mathcal{R}_\infty^{\mathrm{ICL}}(\theta_L(t)) + \nabla\mathcal{R}_\infty^{ICL}(\theta_\infty(t))\| \tag{20}$$
$$\leq \| - \nabla\mathcal{R}_L^{\mathrm{ICL}}(\theta_L(t)) + \nabla\mathcal{R}_\infty^{\mathrm{ICL}}(\theta_L(t))\| + \beta_\infty^{\mathrm{ICL}}\|\theta_L(t) - \theta_\infty(t)\| \tag{21}$$

where the constant $\beta_\infty^{\mathrm{ICL}}$ denotes the Lipschitz constant of $\nabla\mathcal{R}_\infty^{\mathrm{ICL}}$ over $B_{2\rho}$.

To bound $\| - \nabla\mathcal{R}_L^{\mathrm{ICL}}(\theta_L(t)) + \nabla\mathcal{R}_\infty^{\mathrm{ICL}}(\theta_L(t))\|$, we apply Corollary E.5, so that for any $t \in [0, \tau_L]$:

$$\|\dot{\theta}_L(t) - \dot{\theta}_\infty(t)\| \leq g_2(L) + \beta_\infty^{\mathrm{ICL}}\|\theta_L(t) - \theta_\infty(t)\|.$$

Let $f(t) = \|\theta_L(t) - \theta_\infty(t)\|$, as both dynamics starts from the same initialization, we have

$$f(t) = \|\theta_L(t) - \theta_\infty(t)\| = \left\| \int_0^t \dot{\theta}_L(s) - \dot{\theta}_\infty(s) ds \right\| \leq \int_0^t \|\dot{\theta}_L(s) - \dot{\theta}_\infty(s)\| ds.$$

Plugging the previous control on $\|\dot{\theta}_L(s) - \dot{\theta}_\infty(s)\|$, we obtain for any $t \in [0, \tau_L]$ that

$$f(t) \leq \int_0^t \beta_\infty^{\text{ICL}} f(s) \mathrm{d}s + t g_2(L).$$

The integral form of Gronwall's inequality gives that for any $t \in [0, \tau_L]$

$$f(t) \leq g_2(L) \cdot t \cdot \exp\left(\beta_\infty^{\text{ICL}} t\right). \tag{22}$$

By Assumption 4.2, $g_2(L) \xrightarrow[L \to \infty]{} 0$. Therefore for any $T$, we can choose $L^\star$ large enough so that for any $L \geq L^\star$:

$$g_2(L) \cdot T \cdot \exp\left(\beta_\infty^{\text{ICL}} T\right) \leq \rho. \tag{23}$$

Remark that $\theta_L(\tau_L) \notin B_{2\rho}$ by continuity of $\theta_L$, and therefore $f(\tau_L) > \rho$. Now assume that $\tau_L \leq T$. Therefore, Equations (22) and (23) imply that $f(\tau_L) \leq \rho$ for $L \geq L^\star$. This entails that necessarily, $\tau_L \geq T$ for $L \geq L^\star$. As Equation (22) holds on $[0, \tau_L]$, it holds in particular on $[0, T]$, concluding the proof of Lemma E.1. $\qquad\square$

**Lemma E.2.** *Assume that $\ell$ is 1-smooth and that $\nabla_1 \ell(0,0) = 0$. Assume that Assumption 4.2 $(i)$ and $(iii)$ hold. Then*

$$|\mathcal{R}_L^{\text{ICL}}(U,V) - \mathcal{R}_\infty^{\text{ICL}}(U,V)| \leq 2\mathbb{E}_{\mu,y_q}\left[\|y_q\|^4\right]^{1/4} \sqrt{c_1} \mathbb{E}\left[\sigma_\mu^6 \frac{\ln(L)}{L^{c_2/\sigma_\mu^2}}\right]^{1/2}$$
$$\cdot \left[\left(c_1 \mathbb{E}_\mu\left[\sigma_\mu^{12} \frac{\ln(L)^2}{L^{c_2/\sigma_\mu^2}}\right]\right)^{1/4} + \left(\mathbb{E}_{\mu,z_q}\left[\|T^{U,V}[\mu](z_q)\|^4\right]\right)^{1/4}\right],$$

*where the constants $(c_1, c_2)$ are given by Proposition 3.1, and depend solely on $d, V, U, \mathcal{D}$.*

*In particular, when $\mu$ is $\mathcal{D}$ almost surely a Gaussian distribution,*

$$\mathbb{E}_{\mu,z_q}[\|T^{U,V}[\mu](z_q)\|^4]^{1/4} \leq 2\sqrt{d}\|V\|_{\text{op}}\|U\|_{\text{op}} \mathbb{E}[\sigma_\mu^4]^{1/4}$$

*which can be used in the terms given by Proposition E.2.*

*Proof.* By definition,

$$\mathcal{R}_L^{\text{ICL}}(U,V) = \mathbb{E}_{\mu,z}\left[\mathcal{L}_\mu(T^{U,V}[\hat{\mu}_L])\right]$$
$$= \mathbb{E}_{\mu,z}\mathbb{E}_{(z_{\text{query}}, y_{\text{query}})}\left[\ell(T^{U,V}[\hat{\mu}_L](z_{\text{query}}), y_{\text{query}})\right],$$

so that, using the smoothness of $\ell$ and by writing q instead of query for readability,

$$|\mathcal{R}_L^{\text{ICL}}(U,V) - \mathcal{R}_\infty^{\text{ICL}}(U,V)|$$
$$\leq \mathbb{E}_{\mu,z}\mathbb{E}_{(z_q, y_q)}\left[|\ell(T^{U,V}[\hat{\mu}_L](z_q), y_q) - \ell(T^{U,V}[\mu](z_q), y_q)|\right]$$
$$\leq \mathbb{E}_{\mu,z}\mathbb{E}_{(z_q, y_q)}\left[\max(\|(T^{U,V}[\hat{\mu}_L](z_q), y_q)\|, \|(T^{U,V}[\mu](z_q), y_q)\|) \cdot \|T^{U,V}[\hat{\mu}_L](z_q) - T^{U,V}[\mu](z_q\|\right]$$
$$\leq \mathbb{E}_{\mu,z}\mathbb{E}_{(z_q, y_q)}\left[\|y_q\| \left(\|T^{U,V}[\hat{\mu}_L](z_q)\| + \|T^{U,V}[\mu](z_q)\|\right) \|T^{U,V}[\hat{\mu}_L](z_q) - T^{U,V}[\mu](z_q)\|\right].$$

By Cauchy-Schwarz (applied twice),

$$|\mathcal{R}_L^{\text{ICL}}(U,V) - \mathcal{R}_\infty^{\text{ICL}}(U,V)|$$
$$\leq \mathbb{E}_{\mu,z}\mathbb{E}_{(z_q, y_q)}\left[\|y_q\|^4\right]^{1/4} \mathbb{E}_{\mu,z}\mathbb{E}_{(z_q, y_q)}\left[\left(\|T^{U,V}[\hat{\mu}_L](z_q)\| + \|T^{U,V}[\mu](z_q)\|\right)^4\right]^{1/4}$$
$$\cdot \mathbb{E}_{\mu,z}\mathbb{E}_{(z_q, y_q)}\left[\|T^{U,V}[\hat{\mu}_L](z_q) - T^{U,V}[\mu](z_q)\|^2\right]^{1/2}.$$

In what follows, for the sake of conciseness, we use the simple notation $\mathbb{E}$ to refer to $\mathbb{E}_{\mu,z}\mathbb{E}_{(z_q, y_q)}$. The second term is handled as follows

$$\mathbb{E}\left[\left(\|T^{U,V}[\hat{\mu}_L](z_q)\| + \|T^{U,V}[\mu](z_q)\|\right)^4\right] \leq 8\mathbb{E}\left[\|T^{U,V}[\hat{\mu}_L](z_q)\|^4\right] + 8\mathbb{E}\left[\|T^{U,V}[\mu](z_q)\|^4\right]$$
$$\leq 8\mathbb{E}\left[\|T^{U,V}[\hat{\mu}_L](z_q) - T^{U,V}[\mu](z_q)\|^4\right] + 16\mathbb{E}\left[\|T^{U,V}[\mu](z_q)\|^4\right]$$
$$\leq 8c_1 \mathbb{E}\left[\sigma_\mu^{12} \frac{\ln(L)^2}{L^{c_2/\sigma_\mu^2}}\right] + 16\mathbb{E}\left[\|T^{U,V}[\mu](z_q)\|^4\right]$$

by using Lemma C.9 as we did similarly in the proof of Proposition 3.4. Finally, we control the third term by using Proposition 3.1:

$$\mathbb{E}\left[\|T^{U,V}[\hat{\mu}_L](z_{\mathrm{q}}) - T^{U,V}[\mu](z_{\mathrm{q}})\|^2\right]^{1/2} \le \sqrt{c_1}\mathbb{E}\left[\sigma_\mu^6 \frac{\ln(L)}{L^{c_2/\sigma_\mu^2}}\right]^{1/2}.$$

□

**Corollary E.3.** *Assume that $\ell$ is 1-smooth, $\nabla_1\ell(0,0) = 0$ and that Assumption 4.2 holds. Then for any $(U,V) \in B_{2\rho}$, we have*

$$|\mathcal{R}_L^{\mathrm{ICL}}(U,V) - \mathcal{R}_\infty^{\mathrm{ICL}}(U,V)| \le g_1(L)$$

*where $g_1(L) \xrightarrow[L\to\infty]{} 0$.*

*Proof.* To prove this, we use Proposition E.2 by making explicit dependence in $U, V$ for the constant $c_1$ and $c_2$. To this end, we define

$$g_1(L, U, V) = 2\mathbb{E}_{\mu,y_{\mathrm{q}}}\left[\|y_{\mathrm{q}}\|^4\right]^{1/4} \sqrt{c_1(U,V)}\mathbb{E}\left[\sigma_\mu^6 \frac{\ln(L)}{L^{c_2(U,V)/\sigma_\mu^2}}\right]^{1/2}$$
$$\cdot \left[\left(c_1(U,V)\mathbb{E}_\mu\left[\sigma_\mu^{12}\frac{\ln(L)^2}{L^{c_2(U,V)/\sigma_\mu^2}}\right]\right)^{1/4} + \left(\mathbb{E}_{\mu,z_{\mathrm{q}}}\left[\|T^{U,V}[\mu](z_{\mathrm{q}})\|^4\right]\right)^{1/4}\right].$$

Note that $c_1$ and $c_2$ only depend on $U, V$ through their norm and are therefore uniformly controlled in $B_{2\rho}$, i.e., $\bar{c}_1 = \sup_{(U,V)\in B_{2\rho}} c_1(U,V) < \infty$ and $\bar{c}_2 = \inf_{(U,V)\in B_{2\rho}} c_2(U,V) > 0$, so that

$$g_1(L) := \sup_{(U,V)\in B_{2\rho}} g_1(L, U, V)$$

$$\le 2\mathbb{E}_{\mu,y_{\mathrm{q}}}\left[\|y_{\mathrm{q}}\|^4\right]^{1/4} \sqrt{\bar{c}_1}\mathbb{E}\left[\sigma_\mu^6 \frac{\ln(L)}{L^{\bar{c}/\sigma_\mu^2}}\right]^{1/2} \cdot \left[\left(\bar{c}_1\mathbb{E}_\mu\left[\sigma_\mu^{12}\frac{\ln(L)^2}{L^{\bar{c}_2/\sigma_\mu^2}}\right]\right)^{1/4} + M_\rho^{1/4}\right].$$

Assumption 4.2 then implies that this upper bound goes to 0 as $L \to \infty$. □

**Lemma E.4.** *Assume that $\ell$ is 1-smooth and that $\nabla_1\ell(0,0) = 0$. Assume that Assumption 4.2 $(i)$ and $(iii)$ hold. Then*

$$\|\nabla\mathcal{R}_L^{\mathrm{ICL}}(U,V) - \nabla\mathcal{R}_\infty^{\mathrm{ICL}}(U,V)\|_{2\to2} \le$$
$$2\sqrt{2c_1'}\mathbb{E}_\mu\left[\sigma_\mu^6 \frac{\ln(L)}{L^{c_2'/2\sigma_\mu^2}}\right] \cdot \left(\mathbb{E}_{\mu,z_{\mathrm{q}}}[\|T^{U,V}[\mu](z_{\mathrm{q}})\|^2]^{1/2} + \sqrt{c_1}\mathbb{E}_\mu\left[\sigma_\mu^3 \frac{\sqrt{\ln(L)}}{L^{c_2/2\sigma_\mu^2}}\right]^{1/2} + \mathbb{E}[\|y_{\mathrm{q}}\|_2^2]^{1/2}\right)$$
$$+ \sqrt{c_1}\mathbb{E}_\mu\left[\sigma_\mu^3 \frac{\sqrt{\ln(L)}}{L^{c_2/2\sigma_\mu^2}}\right] \cdot \mathbb{E}_{\mu,z_{\mathrm{q}}}[\|\nabla T^{U,V}[\mu](z_{\mathrm{q}})\|^2]^{1/2},$$

*where the constants $(c_1, c_2)$ and $(c_1', c_2')$ are respectively given by Propositions 3.1 and 3.4, and depend solely on $d, V, U, \mathcal{D}$.*

In particular, when $\mu$ is $\mathcal{D}$ almost surely a Gaussian distribution,

$$\mathbb{E}_{\mu,z_{\mathrm{q}}}[\|T^{U,V}[\mu](z_{\mathrm{q}})\|^2]^{1/2} \le 2\sqrt{d}\|V\|_{\mathrm{op}}\|U\|_{\mathrm{op}}\,\mathbb{E}[\sigma_\mu^2]^{1/2}$$
$$\mathbb{E}_{\mu,z_{\mathrm{q}}}[\|\nabla T^{U,V}[\mu](z_{\mathrm{q}})\|^2]^{1/2} \le 2\sqrt{d}(\|V\|_{\mathrm{op}} + \|U\|_{\mathrm{op}})\,\mathbb{E}[\sigma_\mu^2]^{1/2},$$

*which can be used in the terms given by Proposition E.4.*

*Proof.* By definition,

$$\mathcal{R}_L^{\mathrm{ICL}}(U,V) = \mathbb{E}_{\mu,z}\left[\mathcal{L}_\mu(T^{U,V}[\hat{\mu}_L])\right]$$
$$= \mathbb{E}_{\mu,z}\mathbb{E}_{(z_{\mathrm{query}},y_{\mathrm{query}})}\left[\ell(T^{U,V}[\hat{\mu}_L](z_{\mathrm{query}}), y_{\mathrm{query}})\right]$$

and therefore,

$$\nabla \mathcal{R}_L^{\text{ICL}}(U,V) = \mathbb{E}_{\mu,z}\mathbb{E}_{(z_{\text{query}},y_{\text{query}})}\left[\nabla_{U,V}(T^{U,V}[\hat{\mu}_L](z_{\text{query}}))\nabla_1\ell(T^{U,V}[\hat{\mu}_L](z_{\text{query}}),y_{\text{query}})\right]$$

where $\nabla_{U,V}$ is the Jacobian matrix of $(U,V) \mapsto T^{U,V}[\hat{\mu}_L](z_{\text{query}})$. In what follows, we abbreviate query by a simple q to save space in the equations. Since

$$\begin{aligned}
&\nabla \mathcal{R}_L^{\text{ICL}}(U,V) - \nabla \mathcal{R}_\infty^{\text{ICL}}(U,V) \\
&= \mathbb{E}_{\mu,z}\mathbb{E}_{(z_{\text{q}},y_{\text{q}})}\left[\left(\nabla_{U,V}(T^{U,V}[\hat{\mu}_L](z_{\text{q}})) - \nabla_{U,V}(T^{U,V}[\mu](z_{\text{q}}))\right)\nabla_1\ell(T^{U,V}[\hat{\mu}_L](z_{\text{q}}),y_{\text{q}}) \right. \\
&\qquad\qquad\qquad \left. + \nabla_{U,V}(T^{U,V}[\mu](z_{\text{q}})) \cdot \left(\nabla_1\ell(T^{U,V}[\hat{\mu}_L](z_{\text{q}}),y_{\text{q}}) - \nabla_1\ell(T^{U,V}[\mu](z_{\text{q}}),y_{\text{q}})\right)\right]
\end{aligned}$$

$$\begin{aligned}
&\|\nabla \mathcal{R}_L^{\text{ICL}}(U,V) - \nabla \mathcal{R}_\infty^{\text{ICL}}(U,V)\|_{2\to 2} \\
&\leq \sqrt{\mathbb{E}\left[\|\nabla_{U,V}(T^{U,V}[\hat{\mu}_L](z_{\text{q}})) - \nabla_{U,V}(T^{U,V}[\mu](z_{\text{q}}))\|_{2\to 2}^2\right]} \cdot \sqrt{\mathbb{E}\left[\|\nabla_1\ell(T^{U,V}[\hat{\mu}_L](z_{\text{q}}),y_{\text{q}})\|^2\right]} \\
&\qquad + \sqrt{\mathbb{E}\|\nabla_{U,V}(T^{U,V}[\mu](z_{\text{q}}))\|_{2\to 2}^2} \cdot \sqrt{\mathbb{E}\|\nabla_1\ell(T^{U,V}[\hat{\mu}_L](z_{\text{q}}),y_{\text{q}}) - \nabla_1\ell(T^{U,V}[\mu](z_{\text{q}}),y_{\text{q}})\|^2} \\
&\leq \left(\mathbb{E}\left[\|\nabla_{U,V}(T^{U,V}[\hat{\mu}_L](z_{\text{q}})) - \nabla_{U,V}(T^{U,V}[\mu](z_{\text{q}}))\|_{2\to 2}^2\right]\right)^{1/2} \cdot \left(\mathbb{E}\left[\|\nabla_1\ell(T^{U,V}[\hat{\mu}_L](z_{\text{q}}),y_{\text{q}})\|^2\right]\right)^{1/2} \\
&\qquad + \left(\mathbb{E}\|\nabla_{U,V}(T^{U,V}[\mu](z_{\text{q}}))\|_{2\to 2}^2\right)^{1/2} \cdot \left(\mathbb{E}\left\|T^{U,V}[\hat{\mu}_L](z_{\text{q}}) - T^{U,V}[\mu](z_{\text{q}})\right\|^2\right)^{1/2} \\
&\leq \underbrace{\left(\mathbb{E}\left[\|\nabla_{U,V}(T^{U,V}[\hat{\mu}_L](z_{\text{q}})) - \nabla_{U,V}(T^{U,V}[\mu](z_{\text{q}}))\|_{2\to 2}^2\right]\right)^{1/2}}_{\text{use Proposition 3.4}} \cdot \left(\mathbb{E}\left[2\|T^{U,V}[\hat{\mu}_L](z_{\text{q}})\|^2 + 2\|y_q\|_2^2\right]\right)^{1/2} \\
&\qquad + \left(\mathbb{E}\|\nabla_{U,V}(T^{U,V}[\mu](z_{\text{q}}))\|_{2\to 2}^2\right)^{1/2} \cdot \underbrace{\left(\mathbb{E}\left\|T^{U,V}[\hat{\mu}_L](z_{\text{q}}) - T^{U,V}[\mu](z_{\text{q}})\right\|^2\right)^{1/2}}_{\text{use Proposition 3.1}} \\
&\leq \sqrt{2c_1'}\mathbb{E}_{\mu\sim\mathcal{D}}\left[\sigma_\mu^6 \frac{\ln(L)}{L^{c_2'/2\sigma_\mu^2}}\right] \cdot \left(\mathbb{E}\left[2\|T^{U,V}[\hat{\mu}_L](z_{\text{q}})\|^2 + 2\|y_q\|_2^2\right]\right)^{1/2} \\
&\qquad + \sqrt{c_1}\mathbb{E}_{\mu\sim\mathcal{D}}\left[\sigma_\mu^3 \frac{\sqrt{\ln(L)}}{L^{c_2/2\sigma_\mu^2}}\right] \cdot \left(\mathbb{E}\|\nabla_{U,V}(T^{U,V}[\mu](z_{\text{q}}))\|_{2\to 2}^2\right)^{1/2}.
\end{aligned} \tag{24}$$

Using again Proposition 3.1,

$$\mathbb{E}\left[2\|T^{U,V}[\hat{\mu}_L](z_{\text{q}})\|^2 + 2\|y_{\text{q}}\|_2^2\right] \leq 4\mathbb{E}\left[\left\|T^{U,V}[\mu](z_q)\right\|^2\right] + 4\mathbb{E}_\mu\left[c_1\sigma_\mu^6 \frac{\ln(L)}{L^{c_2/\sigma_\mu^2}}\right] + 4\mathbb{E}[\|y_q\|_2^2],$$

which allows to conclude. $\qquad\square$

**Corollary E.5.** *Assume that $\ell$ is 1-smooth, $\nabla_1\ell(0,0) = 0$ and that Assumption 4.2 holds. Then for any $(U,V) \in B_{2\rho}$, we have*

$$|\nabla \mathcal{R}_L^{\text{ICL}}(U,V) - \nabla \mathcal{R}_\infty^{\text{ICL}}(U,V)| \leq g_2(L)$$

*where $g_2(L) \xrightarrow[L\to\infty]{} 0$.*

*Proof.* We use Proposition E.4, and similarly to the proof of Proposition E.5, we can define $\bar{c}_1' = \sup_{(U,V)\in B_{2\rho}} c_1'(U,V) < \infty$ and $\bar{c}_2' = \inf_{(U,V)\in B_{2\rho}} c_2'(U,V) > 0$, so that

$$\begin{aligned}
g_2(L) &\leq \sqrt{\bar{c}_1}\mathbb{E}_\mu\left[\sigma_\mu^3 \frac{\sqrt{\ln(L)}}{L^{\bar{c}_2/2\sigma_\mu^2}}\right] \cdot M_\rho \\
&\qquad + 2\sqrt{2\bar{c}_1'}\mathbb{E}_\mu\left[\sigma_\mu^6 \frac{\ln(L)}{L^{\bar{c}_2'/2\sigma_\mu^2}}\right] \cdot \left(M_\rho + \sqrt{\bar{c}_1}\mathbb{E}_\mu\left[\sigma_\mu^3 \frac{\sqrt{\ln(L)}}{L^{\bar{c}_2/2\sigma_\mu^2}}\right]^{1/2} + \mathbb{E}[\|y_{\text{q}}\|_2^2]^{1/2}\right).
\end{aligned}$$

$\square$

# F. Proofs of Section 5

## F.1. Preliminaries

**Lemma F.1.** *The linear in-context model* (7) *with* $\|\Sigma\|_{\mathrm{op}} \leq 1$ *satisfies Assumption* 4.2.

*Proof.*

**(i).** As $x_{\mathrm{query}}$ is a centered Gaussian vector of covariance matrix $\Sigma$, such that $\|\Sigma\|_{\mathrm{op}} \leq 1$, $x_{\mathrm{query}}$ is 1 sub-Gaussian.

Moreover, as $\mu = \mathcal{N}(0, \Gamma_w)$ where $\Gamma_w = \begin{pmatrix} \Sigma & \Sigma w \\ (\Sigma w)^\top & \|w\|_2^2 \end{pmatrix}$, $\mu$ is $\sigma_\mu$ sub-Gaussian with $\sigma_\mu^2 = \max(\|\Sigma\|_{\mathrm{op}}, \|w\|_2^2)$. In particular, $\mu$ corresponds to a $\max(1, \sigma_\mu)$ sub-Gaussian measure.

**(ii).** Since $\sigma_\mu^2 = \max(\|\Sigma\|_{\mathrm{op}}, \|w\|_2^2)$ and $\|w\|_2^2$ follows a $\chi^2(d)$ distribution, the moment assumption is satisfied, i.e., $\mathbb{E}[\sigma_\mu^{12}] < \infty$.

Moreover, for any $c > 0$ and defining $\Phi : x \mapsto \frac{1}{x} + \frac{x}{2}$,

$$
\begin{aligned}
\mathbb{E}_{\mu \sim \mathcal{D}}\left[\frac{1}{L^{c/\sigma_\mu^2}}\right] &= \mathbb{E}_{\zeta \sim \chi^2(d)}\left[\frac{1}{(L^c)^{1/\max(1,\zeta)}}\right] \\
&\leq \frac{1}{L^c} + \frac{1}{2^{d/2}\Gamma(d/2)}\int_1^\infty x^{d/2-1}\frac{e^{-x/2}}{(L^c)^{1/x}}\mathrm{d}x \\
&\leq \frac{1}{L^c} + \frac{1}{2^{d/2}\Gamma(d/2)}\int_1^\infty x^{d/2-1}\exp\left(-\frac{c\ln(L)}{x} - x/2\right)\mathrm{d}x \\
&\leq \frac{1}{L^c} + \frac{1}{2^{d/2}\Gamma(d/2)}\int_1^\infty x^{d/2-1}\exp\left(-\sqrt{c\ln(L)}\Phi(x/\sqrt{c\ln(L)})\right)\mathrm{d}x \\
&\leq \frac{1}{L^c} + \frac{(c\ln(L))^{d/4}}{2^{d/2}\Gamma(d/2)}\int_0^\infty u^{d/2-1}\exp\left(-\sqrt{c\ln(L)}\Phi(u)\right)\mathrm{d}u.
\end{aligned}
$$

Now, Laplace integral method (see e.g. Ivrii, 2024, Section 2.1, Theorem 2) directly yields the asymptotic equivalence for $u_0 = \arg\min_{u \in \mathbb{R}_+} \Phi(u) = \sqrt{2}$:

$$
\int_0^\infty u^{d/2-1}\exp\left(-\sqrt{c\ln(L)}\Phi(u)\right)\mathrm{d}u \underset{L\to\infty}{\sim} (c\ln(L))^{-1/4}\frac{\sqrt{2\pi}}{\sqrt{\Phi''(u_0)}}u_0^{d/2-1}e^{-\sqrt{c\ln(L)}\Phi(u_0)}.
$$

This equivalence finally yields that for any $c > 0$, $\mathbb{E}_{\mu \sim \mathcal{D}}\left[\frac{\ln(L)^4}{L^{c/\sigma_\mu^2}}\right] \xrightarrow[L\to\infty]{} 0$.

**(iii).** Let $U \in \mathbb{R}^{d \times d}$ and $\mu = \mathcal{N}(0, \Gamma)$ for some matrix $\Gamma$. Similarly to the proof of Lemma 2.1 (in Appendix B.1), we define for any $z = (x_{\mathrm{query}}, 0)^\top$ the skewed distribution $\mu_z$ by:

$$
\mathrm{d}\mu_z(z') = \frac{\exp(z'^\top U z)}{\mathbb{E}_{z_0 \sim \mu}[\exp(z_0^\top U z)]}\mathrm{d}\mu. \tag{25}
$$

And similar computations yield $\mu_z = \mathcal{N}(\Gamma U z, \Gamma)$. Using this relation, we can now bound the following moment for any

$p \in \mathbb{N}$:

$$\frac{\mathbb{E}_{z_1 \sim \mu}[\exp(z_1^\top U z) \|z_1\|^p]}{\mathbb{E}_{z_1 \sim \mu}[\exp(z_1^\top U z)]} = \mathbb{E}_{z_1 \sim \mu_z}[\|z_1\|^p]$$

$$= \mathbb{E}_{Y \sim \mathcal{N}(0,\mathrm{Id})}\left[\|\Gamma U z + \Gamma^{1/2} Y\|^p\right]$$

$$\leq 2^{p-1}\left(\|\Gamma U z\|^p + \mathbb{E}_{Y \sim \mathcal{N}(0,\mathrm{Id})}\left[\|\Gamma^{1/2} Y\|^p\right]\right)$$

$$\leq 2^{p-1}\left(\|\Gamma U z\|^p + \|\Gamma\|_{\mathrm{op}}^{p/2}\mathbb{E}_{Y \sim \mathcal{N}(0,\mathrm{Id})}\left[\|Y\|^p\right]\right)$$

$$\leq 2^{p-1}\max(1,\|\Gamma\|_{\mathrm{op}}^p)\left(\|U\|_{\mathrm{op}}^p\|z\|^p + \underbrace{\mathbb{E}_{Y \sim \mathcal{N}(0,\mathrm{Id})}\left[\|Y\|^p\right]}_{C_{d,p}}\right)$$

for some constants $C_p$ depending solely on $d$ and $p$, using typical moment bounds on multivariate Gaussian distributions. Next, we have $\mathcal{D}$-almost surely that

$$\mathbb{E}_{(x_{\mathrm{query}},y_{\mathrm{query}}) \sim \mu}\left[\frac{\mathbb{E}_{z_1 \sim \mu}[\exp(z_1^\top U(x_{\mathrm{query}},0)^\top)\|z_1\|^p\|x_{\mathrm{query}}\|^q]}{\mathbb{E}_{z_1 \sim \mu}[\exp(z_1^\top U(x_{\mathrm{query}},0)^\top)]}\right]$$

$$\leq 2^{p-1}\sigma_\mu^{2p}\left(\|U\|_{\mathrm{op}}^p\mathbb{E}[\|x_{\mathrm{query}}\|^{p+q}] + C_{d,p}\mathbb{E}\left[\|x_{\mathrm{query}}\|^q\right]\right),$$

which directly yields $(iii)$. $\qquad\square$

### F.2. Proof of Theorem 5.2

The proof of Theorem 5.2 relies on the application of Theorem 4.3 in the specific case of linear in-context setting studied by Zhang et al. (2024).

**Softmax and linear attention in infinite regime.** First of all, we ensure that, in the infinite prompt length limit, the linear attention setting of Zhang et al. (2024) and the softmax attention model described in Equation (7) correspond. Indeed, since we are considering Gaussian data distributions $\mu = \mathcal{N}(0,\Gamma_w)$, Lemma 2.1 directly implies that with softmax attention: $T^{U,V}[\mu](z) = V\Gamma_w U z$.

On the other hand, the linear attention setting of Zhang et al. (2024, Equation (3.6)) corresponds for prompt length $L$, with our notation, to an output

$$T_{\mathrm{lin}}^{U,V}[\hat{\mu}_L]((x,0)^\top) = V\hat{\Gamma}_{L,w,x}U(x,0)^\top$$

where, following our notations, $\hat{\mu}_L = \frac{1}{L}\sum_{k=1}^L \delta_{z_k}$ with $z_k = (x_k, w^\top x_k)$, and such that

$$\hat{\Gamma}_{L,w,x} = \frac{1}{L}\begin{pmatrix} \frac{1}{L}\sum_{k=1}^L x_k x_k^\top + \frac{1}{L}xx^\top & \frac{1}{L}\sum_{k=1}^L x_k x_k^\top w \\ \frac{1}{L}\sum_{k=1}^L x_k^\top wx^\top & \frac{1}{L}\sum_{k=1}^L w^\top x_k x_k^\top w \end{pmatrix}.$$

From here, it is clear that $\hat{\Gamma}_{L,w,x} \xrightarrow[L \to \infty]{} \Gamma_w$ almost surely, so that we indeed have in the infinite case $T_{\mathrm{lin}}^{U,V}[\mu] = T^{U,V}[\mu]$, i.e., in the infinite prompt limit, our softmax attention model does coincide with the linear one described by Zhang et al. (2024), independently of whether the autocorrelation term is included as its contribution vanishes when $L \to +\infty$.

To compare infinite and finite length prompts, we must now ensure that all the required assumptions for Theorem 4.3 are satisfied.

**Verifying the set of assumptions.** First, observe that the quadratic loss is 1-smooth, such that $\nabla_1 \ell(0,0) = 0$. In addition, $\mathcal{R}_\infty^{\mathrm{ICL}}$ is $C^2$, which follows from differentiation under the summation sign.

Then, we have to ensure that the gradient flow trajectory $\begin{pmatrix} U_\infty(t) \\ V_\infty(t) \end{pmatrix}$ for an infinite prompt remains in a ball of a certain radius $\rho$. This fact follows from Theorem 4.1 in Zhang et al. (2024), up to a few minor subtleties. Specifically, they

establish that for a *finite* prompt length, with *linear* attention, the attention parameters satisfy

$$U(t) \xrightarrow[t\to\infty]{} \operatorname{tr}(\Sigma_L^{-2})^{-1/4} \begin{pmatrix} \Sigma_L^{-1} & \mathbf{0}_d \\ \mathbf{0}_d^\top & 0 \end{pmatrix} \quad \text{and} \quad V(t) \xrightarrow[t\to\infty]{} \operatorname{tr}(\Sigma_L^{-2})^{1/4} \begin{pmatrix} \mathbf{0}_{d\times d} & \mathbf{0}_d \\ \mathbf{0}_d^\top & 1 \end{pmatrix},$$

where $\Sigma_L = (1 + \frac{1}{L})\Sigma + \frac{1}{L}\operatorname{tr}(\Sigma)\mathrm{I}_d$. Their analysis extends directly to the infinite-length case $L = \infty$ when $\Sigma$ is invertible; in particular, Equation (5.4) of Zhang et al. (2024) carries over to this limit. The remainder of the argument then follows exactly as in their finite-$L$ setting, yielding the convergence of the gradient flow with linear attention,

$$U_\infty(t) \xrightarrow[t\to\infty]{} \underbrace{\operatorname{tr}(\Sigma^{-2})^{-1/4} \begin{pmatrix} \Sigma^{-1} & \mathbf{0}_d \\ \mathbf{0}_d^\top & 0 \end{pmatrix}}_{=:U^\star} \quad \text{and} \quad V_\infty(t) \xrightarrow[t\to\infty]{} \underbrace{\operatorname{tr}(\Sigma^{-2})^{1/4} \begin{pmatrix} \mathbf{0}_{d\times d} & \mathbf{0}_d \\ \mathbf{0}_d^\top & 1 \end{pmatrix}}_{=:V^\star}.$$

This convergence also holds with softmax attention, since both linear and softmax attention coincide with infinite prompt lengths (see above). The convergence of $\begin{pmatrix} U_\infty(t) \\ V_\infty(t) \end{pmatrix}$ ensures that the trajectory can be confined to a ball of radius $\rho$, for an appropriately chosen $\rho > 0$.

The last remaining step is to check Assumption 4.2; this is precisely ensured by Lemma F.1.

**Convergence guarantees and Bayes optimality.** We can thus apply Theorem 4.3 to in-context linear model (7) so that for any $\varepsilon > 0$, there exists $L(\varepsilon)$ such that for any $L \geq L(\varepsilon)$,

$$\lim_{t\to\infty} \mathcal{R}_L^{\mathrm{ICL}}(U_L(t), V_L(t)) \leq \lim_{t\to\infty} \mathcal{R}_\infty^{\mathrm{ICL}}(U_\infty(t), V_\infty(t)) + \varepsilon.$$

Then, infinite prompt length extension of Theorem 4.1 by Zhang et al. (2024) with our choice of initialization imply that $\lim_{t\to\infty} \mathcal{R}_\infty^{\mathrm{ICL}}(U_\infty(t), V_\infty(t)) = \mathcal{R}(U^\star, V^\star)$. Moreover, Theorem 4.2 by Zhang et al. (2024) ensures that $(U^\star, V^\star)$ admits a theoretical risk of $0$, being therefore Bayes optimal.

### F.3. Extending Zhang et al. (2024) to degenerate covariance $\Sigma$

While the results of Zhang et al. (2024) are restricted to a positive definite covariance matrix $\Sigma$, they can actually be extended to degenerate matrices $\Sigma$, even in the limit case where $L = \infty$.

In that case, the main modifications in their proof will be as follows, **following their notations**.

1) In the limit case $L = \infty$, we first have $\Gamma = \Lambda$.

2) Their Lemma 5.3 will now imply that for any global minimum of $\tilde{\ell}$, we have

$$u_{-1}\Lambda U_{11}\Lambda = \Lambda.$$

3) The PL inequality of Lemma 5.4 would now holds with

$$\mu = \frac{\sigma^2}{\sqrt{d}\|\Lambda\|_{\mathrm{op}}\operatorname{tr}((\Lambda^\dagger)^2)\operatorname{tr}(\Lambda^\dagger)},$$

where $\Lambda^\dagger$ denotes the Moore-Penrose inverse of $\Lambda$.

4) The modified above lemmas are not sufficient anymore to characterize the convergence of $U_1 1$ and $u_{-1}$. Indeed, the set of global minima is now a continuous set. However, using the expression of the time derivative of $U_{11}$ (see in the proof of Lemma A.1), one can note that for any time $t$,

$$U_{11}(t) \in \{\sigma\Theta\Theta^\top + \Sigma M\Sigma \mid M \in \mathbb{R}^{d\times d}\}.$$

This set intersects the set $\{M \in \mathbb{R}^{d\times d} \mid \Lambda M\Lambda = \alpha\Lambda \text{ for some } \alpha > 0\}$ on the affine space given by:

$$\{\alpha\Lambda^\dagger + \sigma P(\Theta\Theta^\top)\},$$

where $P : M \mapsto M - \Lambda\Lambda^\dagger M \Lambda\Lambda^\dagger$ is the orthogonal projection on $\{M \mid \Lambda M \Lambda = \mathbf{0}_{d \times d}\}$. Finally, by balancedness, the model parameters necessarily converge as

$$\lim_{t \to \infty} U_{11}(t) = \alpha \Lambda^\dagger + \sigma P(\Theta\Theta^\top) \quad \text{and} \quad \lim_{t \to \infty} u_{-1}(t) = 1/\alpha,$$

where $\alpha$ ensures balancedness and is given by

$$\alpha = \left( \frac{\sqrt{\sigma^4 \|P(\Theta\Theta^\top)\|_F^4 + 4\mathrm{tr}\left((\Lambda^\dagger)^2\right)} - \sigma^2 \|P(\Theta\Theta^\top)\|_F^2}{2\mathrm{tr}\left((\Lambda^\dagger)^2\right)} \right)^{1/2}.$$

**Going back to our notations,** we would have for degenerate covariance matrix $\Sigma$ that

$$U_\infty(t) \xrightarrow[t \to \infty]{} \begin{pmatrix} \gamma\Sigma^\dagger + \alpha P(\Theta\Theta^\top) & \mathbf{0}_d \\ \mathbf{0}_d^\top & 0 \end{pmatrix} \quad \text{and} \quad V_\infty(t) \xrightarrow[t \to \infty]{} \frac{1}{\gamma} \begin{pmatrix} \mathbf{0}_{d \times d} & \mathbf{0}_d \\ \mathbf{0}_d^\top & 1 \end{pmatrix}$$

with

$$P : M \mapsto M - \Sigma\Sigma^\dagger M \Sigma\Sigma^\dagger$$

$$\text{and} \quad \gamma = \left( \frac{\sqrt{\alpha^4 \|P(\Theta\Theta^\top)\|_F^4 + 4\mathrm{tr}\left((\Sigma^\dagger)^2\right)} - \alpha^2 \|P(\Theta\Theta^\top)\|_F^2}{2\mathrm{tr}\left((\Sigma^\dagger)^2\right)} \right)^{1/2}.$$

In particular, the trajectory of $(U_\infty(t), V_\infty(t))$ remains bounded in that case, so that Theorem 5.2 still holds and still implies a Bayes optimal risk as $L \to \infty$.

