# OpenReview forum: "Softmax as Linear Attention in the Large-Prompt Regime: a Measure-based Perspective"
_ICML.cc/2026/Conference — ICML 2026 regular_

### Official Review · Reviewer_FLQw · 2026-02-22

**Soundness:** 3
**Presentation:** 3
**Significance:** 3
**Originality:** 3
**Overall Recommendation:** 5
**Confidence:** 3

**Summary:**

This paper develops a measure-based framework for studying one-layer softmax attention with both finite and infinite prompts. It discovers a quantitative bounds which is related to the concentration of finite-prompts to the infinite-prompts. Then, the authors show that this concentration persists in the general in-context-learning settings. At last, this paper applies the results to in-context linear regression task to show the potential application of the results and correctness of the results empirically.

**Compliance With Llm Reviewing Policy:**

Affirmed.

**Final Justification:**

This paper is well-constructed, and technically solid. The rebuttal addressed my main concerns, and I keep my positive review with the score at 5.

**Key Questions For Authors:**

- The initialization of the softmax attention is based on a specific scheme. Could authors provide more specific reasons about this choice? How random initialization can affect the analysis of training dynamics? Because in [1], they use the random initialization for the attention.
- The results for concentration in Section 3 are not related to the specific covariance of the input sequence z_\ell. However, in the experiments, authors only use the simplest isotropic covariates. Could authors investigate some complex scenarios. For example, setting \Sigma=U^\top\Lambda U, where U is the rotation matrix and \Lambda is a diagonal matrix with the d eigenvalues.

Reference: [1] In-context linear regression demystified: Training dynamics and mechanistic interpretability of multi-head softmax attention.

**Limitations:**

This paper mainly focus on the single layer attention, while modern transformers derive their expressivity from deep stacks of attention layers. It is a potential limitation that this framework does not include dynamics of multi-layer softmax attentions, although it will be more complex.

**Strengths And Weaknesses:**

**Strength:**
- The paper is well-constructed and provides the clear, comprehensive proofs and explanations about the proposed framework. The framework is novel and successfully unifies the study of single-layer softmax attention under both finite and infinite-prompt settings.
- This paper studies from a big picture, demonstrating that this concentration remains stable across the entire training trajectory for general in-context learning tasks involving sub-Gaussian tokens.
- The experiments clearly demonstrate that the outputs, gradients, and overall training trajectories of finite softmax models converge with their linear counterparts as prompt lengths increase, confirming the core theoretical claims.

**Weakness:**
- While mathematically convenient, the specific block-matrix initialization used for the attention parameters may limit how well these empirical results generalize to practical, real-world models.
- The experiments in this paper are in limited scenarios, which may limit the generalizability of the findings.

---

> ### Author Rebuttal · Authors · 2026-03-30
>
> We thank the reviewer for his/her comments and we provide specific answer in the following.
>
> **Q1.** In Section 5, our goal is to apply the general framework developed in the previous sections to the result of Zhang et al. (2024), in order to derive training dynamics results for in-context learning with softmax attention. For this reason, we adopt their initialization scheme.
> In their original work, this block-structured initialization plays an important role in making the analysis tractable, as studying training dynamics remains challenging even in the linear attention setting. This choice allows them to isolate the relevant mechanisms and obtain clean theoretical results.
>
> That said, this assumption is primarily a technical device and does not appear to be necessary in practice. This is supported by our experiments, where we consider standard random initializations and observe qualitatively similar behaviors.
> While He et al. (2025) consider more general initialization schemes, they do not provide rigorous results on training dynamics—their Section 4.2 remains heuristic and does not include formally proved guarantees.
> We also note that this type of block-structured initialization is not specific to Zhang et al. (2024), but has been adopted in other works, such as Chen et al. (2025), where it is extended to the multi-head setting (see their Definition 3.1).
>
> **Q2.** While the experiments in the main paper focus on isotropic Gaussian data, we also report in Appendix A.2.2 additional experiments with anisotropic covariance structures. In this setting, the covariance is generated as a Kac–Murdock–Szegő matrix, following a similar setup to the anisotropic experiments in He et al. (2025). Although the convergence to the infinite-prompt limit is somewhat slower than in the isotropic case, we observe qualitatively similar behaviors.

---

> > ### Author Rebuttal · Reviewer_FLQw · 2026-04-01
> >
> > Thanks for authors' response. My questions are fully resolved. I keep my score at 5.

---

### Official Review · Reviewer_UtS5 · 2026-03-05

**Soundness:** 3
**Presentation:** 4
**Significance:** 3
**Originality:** 3
**Overall Recommendation:** 5
**Confidence:** 4

**Summary:**

The paper reformalizes the attention equation under a measure perspective. It first states concentration properties of softmax attention and its derivatives as prompt length goes to infinity, then turns to in-context learning and proves the closeness of ICL risk between infinite and finite length prompt under both general and linear regression settings. The empirical results show that softmax attention behaves similarly to linear attention as prompt length $L$ gets larger.

**Compliance With Llm Reviewing Policy:**

Affirmed.

**Final Justification:**

The authors rebuttal clarifies the relation between Lemma 2.1 and a standard LSA, and promises to add a clear clarification. The takeaway of the paper is interesting and is worth accepting from my perspective.

**Key Questions For Authors:**

1. Could you clarify the difference of Lemma 2.1 and a standard LSA? I feel it would be better if the gap could be closed.
2. How might the results change when Gaussian assumptions don't hold? Will it be possible to analyze under a more general data distribution, or extend the setting of linear regression to classification tasks etc.?
3. Could you write Theorem 4.1 and 5.3 in an asymptotic way, e.g. explicitly write $\epsilon = \epsilon (L)$ so that the convergence rate is clear?

**Limitations:**

yes

**Strengths And Weaknesses:**

Strengths:
1. The writing, presentation and story telling is clear and easy to follow.
2. It is clever to consider the measure-based perspective in the ICL setting, because the softmax attention $\rightarrow$ LSA property only holds for i.i.d. distribution of tokens (prompts), which inherently coincides with ICL.
3. The takeaway that the behavior of softmax attention converges to LSA is interesting and justifies the choice of LSA by many related works.

Weaknesses:
1. Theorems 4.1 and 5.3 require $L$ large but give no explicit “how large”.
2. Lemma 2.1 shows softmax attention is approximately linear, but this is not the same format as LSA.

---

> ### Author Rebuttal · Authors · 2026-03-30
>
> We thank the reviewer for his/her comments and we provide specific answer in the following.
>
> ## Questions
>
> **Q1.** (Normalized) standard LSA is given by: $$ \frac{1}{L}\sum_{\ell = 1}^L \langle Q z_q, K z_{\ell}\rangle V z_{\ell}.$$
> In contrast to much of the literature, we include the normalization factor $\frac{1}{L}$ so that the limit as $L \to \infty$ is well-defined.
>
> A straightforward application of the law of large numbers then yields the pointwise limit for $T[\mu] (z_q)$ as
> $$V \mathbb{E}_{z\sim\mu}[z z^\top]K^\top Q z_q.$$
> This expression coincides with the formula in Lemma 2.1 in the special case of **centered** Gaussian inputs. Accordingly, the equivalence between softmax and linear attention holds only under this centering assumption, which is standard in much of the linear attention literature.
>
> For non-centered Gaussian tokens, softmax attention is no longer equivalent to linear attention in the $L \to \infty$ limit. Nevertheless, it still defines a linear (more precisely, affine) transformation, both with respect to the model parameters and to the underlying input measure.
> We will include this clarification and a more detailed comparison with linear attention in the revised version.
>
> **Q2.** The results established in Section 3 are not specific to Gaussian distributions, but hold more generally under sub-Gaussian assumptions. That said, for general sub-Gaussian inputs, the limiting transformation (as $L \to \infty$) does not admit as simple a structure as in the Gaussian case, where it reduces to a linear map. Nevertheless, we expect that for many data distributions of practical interest, the limit transformation retains a sufficiently structured form to allow for tractable analysis of training dynamics, which is the focus of current on-going works.
>
> Regarding classification tasks, Theorem 4.3 is not specific to regression and applies equally to standard classification losses such as cross-entropy. One subtlety, however, is that in classification settings the parameters may diverge as $t \to \infty$ in the infinite-prompt regime, in which case the boundedness assumption required in Theorem 4.3 may fail. This phenomenon typically arises in separable regimes; in contrast, when considering population-level objectives or settings with label noise, such divergence is generally avoided. Therefore, we emphasize that Theorem 4.3 remains broadly applicable to classification, and we believe it provides a useful foundation for analyzing such settings.
>
>
> **Q3.** The convergence rates in our results also depend on those inherited from the infinite-prompt dynamics. That said, our analysis is not designed to yield optimal rates, but rather to capture the limiting behavior as $L \to \infty$ in a general setting. Obtaining sharper rates—particularly in the setting of Section 5—would likely require additional structural assumptions, that are more specific to the considered problem (e.g., a local Polyak–Łojasiewicz condition for the infinite-prompt dynamics).
>
> We will also clarify in the main text that the primary goal of our framework is not to derive optimal convergence rates, but rather to characterize the limiting behavior of the dynamics as L becomes large. Obtaining sharp rates typically requires more specialized and technically involved arguments, tailored to the specific problem and not relying on the infinite-prompt perspective. As discussed in Section 5, this approach has been pursued by Chen et al. (2025) in the context of ICL with softmax attention, yielding tight rates at the cost of more intricate, problem-specific analysis and stronger assumptions, such as isotropic covariance of the token distribution.

---

> > ### Author Rebuttal · Reviewer_UtS5 · 2026-04-03
> >
> > I thank the authors for the rebuttal, and I will keep my positive review of this paper.

---

### Official Review · Reviewer_rQVZ · 2026-03-13

**Soundness:** 3
**Presentation:** 2
**Significance:** 3
**Originality:** 3
**Overall Recommendation:** 5
**Confidence:** 4

**Summary:**

Using a  measure-based formulation of attention, this paper shows that under sub-gaussian assumptions on the input sequence, a one layer softmax attention model behaves similarly on a finite number of samples than on an infinite number of samples as this number of samples goes to infinity, both in terms of output values and gradients under appropriate reparametrization of key and query projections. This enables to leverage a known result by Castin et al. (2025) showing that softmax attention behaves like linear attention (i.e. where softmax is replaced by identity)  on a gaussian measure to generalize the result of Zhang et al. 2024 (showing that one layer transformers provably learn the Bayes optimal  predictor in linear regression in-context learning setup) to softmax attention. Numerical results are presented to validate the theoretical claims.

**Compliance With Llm Reviewing Policy:**

Affirmed.

**Final Justification:**

The authors adresed all my concerns and I am therefore increasing my score from 4 to 5.

**Key Questions For Authors:**

These are more remarks that I would like to be addressed
l.37: it looks like you are claiming you are the one introducing this framework.

l.52: same you make it appear as a contribution which is unfair

l. 98: the litterature review is not well done. Please update this part with relevant references to how attention on measure was introduced. More time should be spent here, it really looks like when reading your paper that you are the one introducing the concept. The particle-based perspective was not first developed by Geshkovski et al.

l. 106 same you are not the first to restrict to a single layer, see for instance Castin et al. 2023.

l.92 (outline part) you do not introduce anything, you just reintroduce.

Section 2: it is presented as if you introduce this concept yourself.

l. 162 this is the result from Castin et al. and could be mentionned directly in the name of the lemma.

l.228 how realistic is this assumption?

l.304: please give more refs, this is widely studied.

eq. (9): make it an assumption?

Figure 1: make it in log scale?

**Limitations:**

yes

**Strengths And Weaknesses:**

The submission is technically very sound and the theoretical results presented are strong and impactful for the community. The experimental results could be better presented.

The submission could benefit from being more pedagogical. I found the narrative between the general case and the in-context learning hard to follow. In general the paper is very dense and I would advise for adding clarity, at the cost of moving some things to the appendix.    The main criticism I have is regarding how the work positions itself in the context of prior literature. The related work section is incomplete and the relevant works are not cited. I advise the authors to carefully amend this part of the paper by doing the necessary bibliography work. Happy to help complete this part once the authors make a first proposition in their response.


I believe the paper adresses an important and relevant problem: a large body of works study this measured based interpretation of transformers and this paper quantifies how well considering the case where the measure has a density actually makes sens for finite length prompt. The result on in-context learning is strong and is particularly relevant in the light of large scale reasoning models being widely used.

The work is original though I believe it should better situate itself with respect to existing works on very related topics, which relate to my remark in the presentation bullet point above. That being said, the theorems are new and not incremental at all and I believe they should be published, after the presentation of the paper is improved which I hope will be the case during the rebuttal phase. Very few references are given.

For now I believe the paper deserves a weak-accept and I may change my rating based on the rebuttal, in particular if my points are addressed or not.

---

> ### Author Rebuttal · Authors · 2026-03-30
>
> We thank the reviewer for their comments, which helped improve the paper. We provide detailed responses to the points and questions raised below.
>
> ## Weaknesses
>
> **W1 (Presentation)** We deliberately kept the framework in Sections 3 and 4 as general as possible, as the concentration tools we introduce are intended to support detailed theoretical analyses of training dynamics—particularly in the setting of in-context learning. In this sense, these sections are meant to serve as a flexible theoretical toolbox that can be reused and adapted in future work. We acknowledge that this level of generality comes with heavier notation, and we would welcome any suggestions from the reviewer to improve clarity.
>
> **W2 (Biblio)** We thank the reviewer for pointing out that we were missing references on measure-based perspectives of attention. After further bibliographical search, we identified the works of Vuckovic et al. (2020), Sander et al. (2022), Bruno et al. (2024), Burger et al. (2025), Zimin et al. (2025), Chen et al. (2025), Rigollet (2025) who also adopted a measure-based view of attention. Of course, we would be glad to add any other reference suggested by the reviewer on that matter.
>
> Although not explicitly mentioning attention, the works of Pevny and Kovarick (2019), De Bie et al. (2019), Zweig and Bruna (2021) also study architectures taking measures as input and will also be cited in the revised version. As raised by the reviewer, we would like to stress that the existence of these previous works does not diminish the novelty of our contribution, since it is the first one (to our knowledge) to leverage this measure-based view to derive training dynamics result with finite prompt-length.
>
> ## Questions.
>
> **l37-l52-l92-l98-Section 2** We apologize for the lack of clarity. We will revise the introduction and the beginning of Section 2 to better emphasize that our construction builds on prior works, such as those of Vuckovic et al., Sander et al., Castin et al.
>
> **l106** Indeed, we will correct this sentence as follows: ``In contrast to prior approaches that focus on the architecture inference through attention layers with fixed weights, we here study the training dynamics of a single attention layer, with large but finite prompt length. '', making the distinction with previous work more accurate.
>
> **l162-eq 9-Fig 1** We thank the reviewer for these suggestions and will include them in the revised version.
>
> **l228 (Assumption 4.2)**
> Item (i) corresponds to assuming sub-Gaussian tokens, a standard and convenient hypothesis in this line of work.
> Item (ii) further requires that the (random) sub-Gaussianity parameter $\sigma_{\mu}$ is itself light-tailed, ensuring that the resulting concentration properties remain well-controlled.
> Item (iii) guarantees that, after passing through the attention layer, the outputs admit sufficiently bounded moments.
>
> Taken together, these assumptions are relatively mild and provide a natural and tractable theoretical framework for the analysis.
>
> This discussion will be added to the revised version.
>
> **l304** In-context linear regression has been indeed widely studied, and we have included several relevant references in the *Related Works* section (lines 55–73). We would be very happy to incorporate additional references suggested by the reviewer to further strengthen this part of the paper.
>
> In line 304, we chose to highlight Zhang et al. (2025) more specifically, as this section builds directly on their framework and adopts their formalism to introduce our setting. This was not meant to downplay other contributions on the topic, but rather to clarify the immediate technical lineage of our approach.
>
>
> ---------------
>
> # References
>
> Vuckovic, J., Baratin, A., and Combes, R. T. D. (2020). A mathematical theory of attention.
>
> Sander, M. E., Ablin, P., Blondel, M., and Peyré, G. (2022, May). Sinkformers: Transformers with doubly stochastic attention.
>
> Bruno, G., Pasqualotto, F., and Agazzi, A. (2024). Emergence of meta-stable clustering in mean-field transformer models.
>
> Burger, M., Kabri, S., Korolev, Y., Roith, T., and Weigand, L. (2025). Analysis of mean-field models arising from self-attention dynamics in transformer architectures with layer normalization.
>
> Zimin, A., Kutakh, A., Polyanskiy, Y., and Rigollet, P. (2025).
>
> Chen, S., Lin, Z., Polyanskiy, Y., and Rigollet, P. (2025). Quantitative clustering in mean-field transformer models.
>
> Rigollet, P. (2025). The mean-field dynamics of transformers.
>
> Pevny, T., and Kovarík, V. (2019). Approximation capability of neural networks on spaces of probability measures and tree-structured domains.
>
> De Bie, G., Peyré, G., and Cuturi, M. (2019, May). Stochastic deep networks.
>
> Zweig, A., and Bruna, J. (2021, July). A functional perspective on learning symmetric functions with neural networks.

---

> > ### Author Rebuttal · Reviewer_rQVZ · 2026-04-03
> >
> > Dear authors
> >
> > Thank you for your rebuttal. I am satisfied with the proposed references. Please do include them in the camera ready version of the paper. Please also leverage the additional 1 page to de-densify the paper. I am increasing my score to 5.
> >
> > Regarding ICL, here a 3 references that you can include:
> >
> > https://arxiv.org/abs/2307.03576
> >
> > https://arxiv.org/abs/2306.09927
> >
> > https://arxiv.org/abs/2306.00297

---

> > > ### Author Response · Authors · 2026-04-03
> > >
> > > Thank you for your positive feedback and for updating your score! We will include the suggested references in the new version and use the additional page to improve clarity.

---

### Decision · Program_Chairs · 2026-04-30

**Decision:**

Accept (regular)

**Comment:**

This paper studies single-layer softmax attention through a measure-based framework. It establishes non-asymptotic concentration bounds and enables the analysis of training dynamics in in-context learning under sub-Gaussian inputs. The work further clarifies the connection between softmax and linear attention in the large-prompt regime.

All reviewers agree that the paper is technically strong and addresses an important theoretical problem. The rebuttal fully resolved their initial concerns, and all reviewers support acceptance.